# Temporal chromatin accessibility changes define transcriptional states essential for osteosarcoma metastasis

W. Dean Pontius [1,2] ✉, Ellen S. Hong[1], Zachary J. Faber [1], Jeremy Gray[1], Craig D. Peacock [1], Ian Bayles[1], Katreya Lovrenert[1], Diana H. Chin [1], Berkley E. Gryder[1], Cynthia F. Bartels[1] & Peter C. Scacheri[1,3] ✉

The metastasis-invasion cascade describes the series of steps required for a cancer cell to successfully spread from its primary tumor and ultimately grow within a secondary organ. Despite metastasis being a dynamic, multistep process, most omics studies to date have focused on comparing primary tumors to the metastatic deposits that define end-stage disease. This static approach means we lack information about the genomic and epigenomic changes that occur during the majority of tumor progression. One particularly understudied phase of tumor progression is metastatic colonization, during which cells must adapt to the new microenvironment of the secondary organ. Through temporal profiling of chromatin accessibility and gene expression in vivo, we identify dynamic changes in the epigenome that occur as osteosarcoma tumors form and grow within the lung microenvironment. Furthermore, we show through paired in vivo and in vitro CRISPR drop-out screens and pharmacological validation that the upstream transcription factors represent a class of metastasis-specific dependency genes. While current models depict lung colonization as a discrete step within the metastatic cascade, our study shows it is a defined trajectory through multiple epigenetic states, revealing new therapeutic opportunities undetectable with standard approaches.

Over 90% of cancer deaths occur due to metastasis, or the spreading of tumor cells from their original location to other sites within the body[1]. While the clinical need to target metastasis is clear, the development of anti-metastatic therapies has proved to be difficult for two main reasons. First, large-scale sequencing studies have failed to find genetic aberrations that drive metastatic progression. Unlike tumorigenesis, whose causal mutations are well characterized in a variety of cancers, metastasis is associated with few recurrent genetic events[2–4]. Second, the spread of cancer cells to a distal site is a complex, multistep process known as the metastasis-

invasion cascade[5]. Not only is the mechanism required for initial dissemination distinct from that required for successful growth and survival at the distal site, but each individual stage may be a complex biological process in itself[6]. For example, the final stage of the metastasis-invasion cascade, known as colonization, encompasses everything from the formation of clinically undetectable micro-metastases to full-blown metastatic disease. During this period of metastasis development, tumor cells are actively adapting to the stresses of a different organ's cellular millieu in order to ultimately grow in an uncontrolled manner. This transitional middle phase may

[1]Department of Genetics and Genome Sciences, Case Western Reserve University School of Medicine, Cleveland, OH, USA. [2]Department of Molecular Medicine, Cleveland Clinic Lerner College of Medicine of Case Western Reserve University, Cleveland, OH, USA. [3]Present address: Amgen Research, Discovery Biomarkers, Thousand Oaks, CA, USA. ✉e-mail: wdp14@case.edu; pcscach@gmail.com

provide a unique clinical opportunity, yet the underlying biology remains largely unexplored.

Cancer cells accumulate genetic and epigenetic changes during metastatic progression. At the level of the enhancer epigenome, these pro-metastatic changes have been characterized by our lab and others across numerous cancer types[7–9]. Despite recognition that these changes occur sometime during the evolution of the tumor, when, why, and how they occur remain open questions. Prior studies have relied on profiling and comparing primary tumors to clinically resectable metastases. By limiting comparison to these endpoints, the natural history of cellular states throughout the metastatic cascade is lost. Additionally, while such an approach may be sufficient to capture inherited and somatic genetic mutations that persist in the final output, epigenetic changes are inherently plastic and context-dependent. Thus, they could easily be missed. This leaves the potential for transient intermediate cell states that are undetectable with these static comparisons. Temporal profiling over a defined time course of metastatic colonization is required to investigate this. Genetically engineered mouse models are ideal systems for studying tumor evolution from a single cell, but the stochastic nature of lesion formation makes isolating defined stages of metastasis development challenging. To chart the landscape of epigenetic changes that occur during metastatic colonization, we need a tractable and controlled experimental system with well-characterized growth dynamics from metastatic seeding to full colonization. Further, a system amenable to functional perturbation across multiple different human cancer specimens, within the in vivo microenvironment, would be ideal.

The lung is the second-most frequent site of cancer metastasis, with twenty-five to thirty percent of all patients with cancer at autopsy having visible lung metastases[10]. This occurs across many primary tumor types. One of these is osteosarcoma—a particularly aggressive bone cancer that is the second leading cause of cancer deaths in adolescents and young adults[11]. Despite its favorable prognosis in patients with localized disease, osteosarcoma that has metastasized portends a five-year survival rate below 30%—a statistic that has not improved in the last four decades.

Here, leveraging osteosarcoma models and functional genomics to study regulatory programs that underlie lung metastasis, we find that colonization is not a single step defined by a single transcriptional program. Instead, it is a trajectory of cell states regulated by distinct factors essential for metastatic progression.

## Results

### Temporal profiling of gene regulation during osteosarcoma metastasis

We performed ATAC-seq and RNA-seq on the metastatic human osteosarcoma cell line MG63.3-GFP grown in vitro, as well as harvested from mouse lung at 1 and 22 days after intravenous inoculation (Fig. 1A). Although this model of metastasis bypasses dissemination from the primary tumor and intravasation into the circulation, the growth dynamics of subsequent metastases are reproducible and well characterized. Fully formed tumors are present after three weeks, with mice succumbing to metastatic disease at one month[7,12,13]. The reproducible nature of this system makes it ideal for studying the regulatory mechanisms underlying colonization. In addition, functional validation is flexible and straightforward due to the ability to perturb gene function in any lung-metastatic human cell line prior to injection. As expected, tumor burden within the lung is visibly different at the two time points assessed, with single cancer cells dispersed throughout the lung at day 1, and larger tumors observed at day 22 (Fig. 1A). We reasoned that profiling these two time points would capture changes associated with both migration/ colonization (early steps) and proliferation/outgrowth within the metastatic organ (late steps). Through principal component analysis of the log2 normalized ATAC-seq profiles, we observed clustering of biological replicates from each of the three conditions (Fig. 1B). This indicated robust and reproducible epigenomic changes at the time points analyzed. The corresponding analysis of the matched transcriptomes showed similar clustering. These data indicate that the regulatory programs of osteosarcoma cells at the early and late stages of lung metastasis are largely distinct.

Regions that change in accessibility during metastatic progression are distributed throughout the genome, with some changes

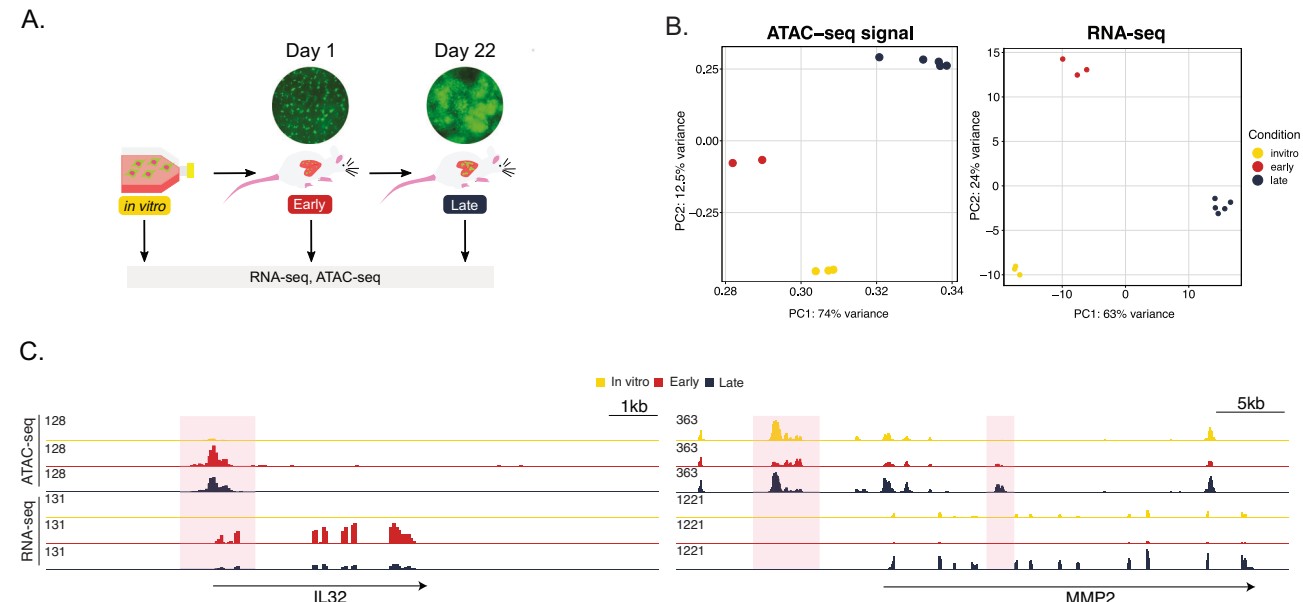

**Fig. 1 | RNA expression and open chromatin are dynamic during osteosarcoma lung metastasis. A** Experimental schematic of gene regulation profiling during lung metastasis. Metastatic osteosarcoma cells were injected intravenously. **B** Principal component analysis of the open chromatin profiles and transcriptomes of cells isolated from three metastasis time points. Each individual point represents a distinct biological replicate. **C** Representative genome-browser screenshots of dynamic accessible regions and corresponding transcripts.

occurring at gene promoters, and others occurring at intergenic or intronic regions (Fig. 1C). Many of the associated genes have been investigated in the context of osteosarcoma metastasis, but underlying mechanisms contributing to their aberrant expression remain unclear. For example, the gene *IL32*, which encodes a cytokine shown to promote osteosarcoma cell invasion and motility, displays an increase in chromatin accessibility at its promoter primarily at the early time point[14] (Fig. 1C). This increase in accessibility is reflected in the gene's expression. In addition, the collagenase encoded by *MMP2* has been associated with osteosarcoma pulmonary metastasis[15]. Our data demonstrate that *MMP2* expression increases at the late in vivo time point, and is associated with an increase in chromatin accessibility at a putative intronic enhancer element (Fig. 1C).

To determine if these findings generalize to other specimens, we repeated the experiment using another metastatic osteosarcoma cell line model: 143b-HOS-GFP (Supplementary Fig. 1). This revealed that dynamic changes in chromatin accessibility also occur in 143b-HOS-GFP, with the three different conditions again clustering separately at both the chromatin and RNA levels (Supplementary Fig. 1A, B). While the exact regions that changed in accessibility were not identical between the two cell lines, there was some overlap (Supplementary Fig. 1C). For example, the promoter of *IL32* again displayed an early-specific increase in accessibility (Supplementary Fig. 1D). This implies common biology underlies the different stages of osteosarcoma metastasis despite genetic and epigenetic variation, and that these programs are regulated by reproducible changes in the epigenomes of metastasizing cells.

## Dynamic shifts in chromatin accessibility during osteosarcoma progression correlate with temporally distinct metastatic programs

We used k-means clustering to partition the dynamic epigenome of MG63.3-GFP cells based on how each region changes in chromatin accessibility over the metastasis time course. We found eight robust clusters (Fig. 2A, B). The first four clusters contained regions more accessible at one or more of the in vivo time points, compared to in vitro. We thus annotated them as early (cluster 1), pan in vivo (clusters 2 & 3), and late (cluster 4), and will refer to them as such throughout the rest of the text. In addition to the clusters of peaks gained in vivo, we also identified clusters that were lost throughout metastasis (cluster 6), or at a specific time during metastatic progression. Cluster 5, for example, contains regions that transiently decrease in accessibility early, while cluster 8 is composed of regions that are relatively inaccessible late.

Positive cis-regulatory elements like enhancers and promoters lie within regions of accessible chromatin[16]. However, accessible chromatin also houses insulator elements that help constrain three-dimensional genomic interactions within confined regions, and are bound by structural transcription factors such as CTCF[16]. If the regions we identified are acting predominantly as functional enhancers or promoters, we would expect the expression of their target genes to change in a similar fashion.

Analysis of the expression of target genes predicted by the Genomics Regions Enrichment of Annotations Tool (GREAT) revealed a general correspondence between accessibility and gene expression changes for clusters 1–4 (Fig. 2C)[17]. We found that genes associated with the early cluster (1) displayed the highest level of expression at the early time point. Similarly, genes paired with the pan in vivo clusters (2 & 3) showed higher expression in both in vivo conditions when compared to in vitro. This same phenomenon held true for the late cluster (4), whose target genes were expressed highest at the late time point. However, correlation between expression and accessibility was weak for the other clusters, potentially due to nuances in the relationship between chromatin accessibility and gene expression. For example, a substantial change in the accessibility of an enhancer may not

correspond to a similarly substantial change in the expression of the linked gene. In addition, accessible enhancers are not always engaged in active gene regulation at the moment of profiling and are instead poised for future activity. Lastly, as mentioned before, these clusters may be enriched for other regulatory elements that do not positively correlate with associated gene expression, such as insulators elements. Despite these intricacies, chromatin accessibility profiles still provide meaningful insights into genes that could play vital roles during metastasis.

We used GREAT to identify biological processes associated with the three in vivo gained clusters (early, pan in vivo, and late; Clusters 1–4) (Fig. 2D)[17]. Biological terms for each cluster were quite distinct, and included processes with clear ties to metastasis. This suggests that the clusters regulate gene programs corresponding to distinct stresses faced throughout metastasis. The early regions (cluster 1) regulate genes associated with fluid shear stress, a likely response to the mechanical forces present during migration through the circulation, and extravasation into the secondary organ. Additionally, this same cluster seems to be responsible for regulating apoptosis and senescence−two pathways critical for survival of micrometastases in the stressful new microenvironment of the lung[18,19]. The pan in vivo regions (clusters 2 & 3) are enriched for terms involving extracellular stimuli, such as response to growth factors and cell-to-cell interactions. These are biological processes required for osteosarcoma cells to interact with the new cell types that make up the milieu of the metastatic microenvironment. Interestingly, cluster 3 shows specific enrichment for many terms involved with lymphocyte and regulatory T cell differentiation in spite of our in vivo model lacking functional lymphocytes. This could imply lung colonization depends on an immune suppressive phenotype mediated by changes in the osteosarcoma epigenome. Lastly, the late regions (cluster 4) associate with pathways important for the growth of larger metastatic lesions. These pathways include remodeling of the local extracellular matrix, activation of bone development programs important for osteosarcoma growth, and activation of vasculogenic programs required to support the increased nutrient and waste transport needs of larger tumors.

To confirm the robustness of these results, we used an ANOVA to determine the number of peaks that were significantly different between time points ($p < 0.05$). This analysis showed 96.6% of peaks used in GREAT were significantly different across time points. Redoing the GREAT analysis with only those peaks yielded the same results, reinforcing the validity of these findings.

Strikingly, when we assessed how expression of the genes associated with these specific GREAT terms changed over our metastatic time course, there was an even closer association between gene expression and chromatin dynamics (Fig. 2E). In the 143b-HOS-GFP cell line, not only do we see partial overlap of the significantly enriched terms for each cluster, but of the specific accessible regions within each cluster as well (Supplementary Fig. 2). These overlaps are statistically significant, indicating that even in tumors with genetically diverse backgrounds, common transcriptional programs are required to successfully navigate the different stages of lung metastasis represented in our model (Supplementary Fig. 2C, D).

## Distinct transcription factors regulate temporal chromatin clusters

Transcription factors mediate the regulation of genes through context-specific interactions with enhancers and promoters[20]. To identify transcription factors controlling the temporally distinct changes in chromatin accessibility, we analyzed the DNA sequence within each cluster of accessible regions to find differentially enriched motifs (Fig. 3A). This analysis revealed unique sets of TFs predicted to bind to the various accessible chromatin clusters. While the early regions were highly enriched for FOX motifs and NFkB motifs, the pan in vivo regions showed specific enrichment of

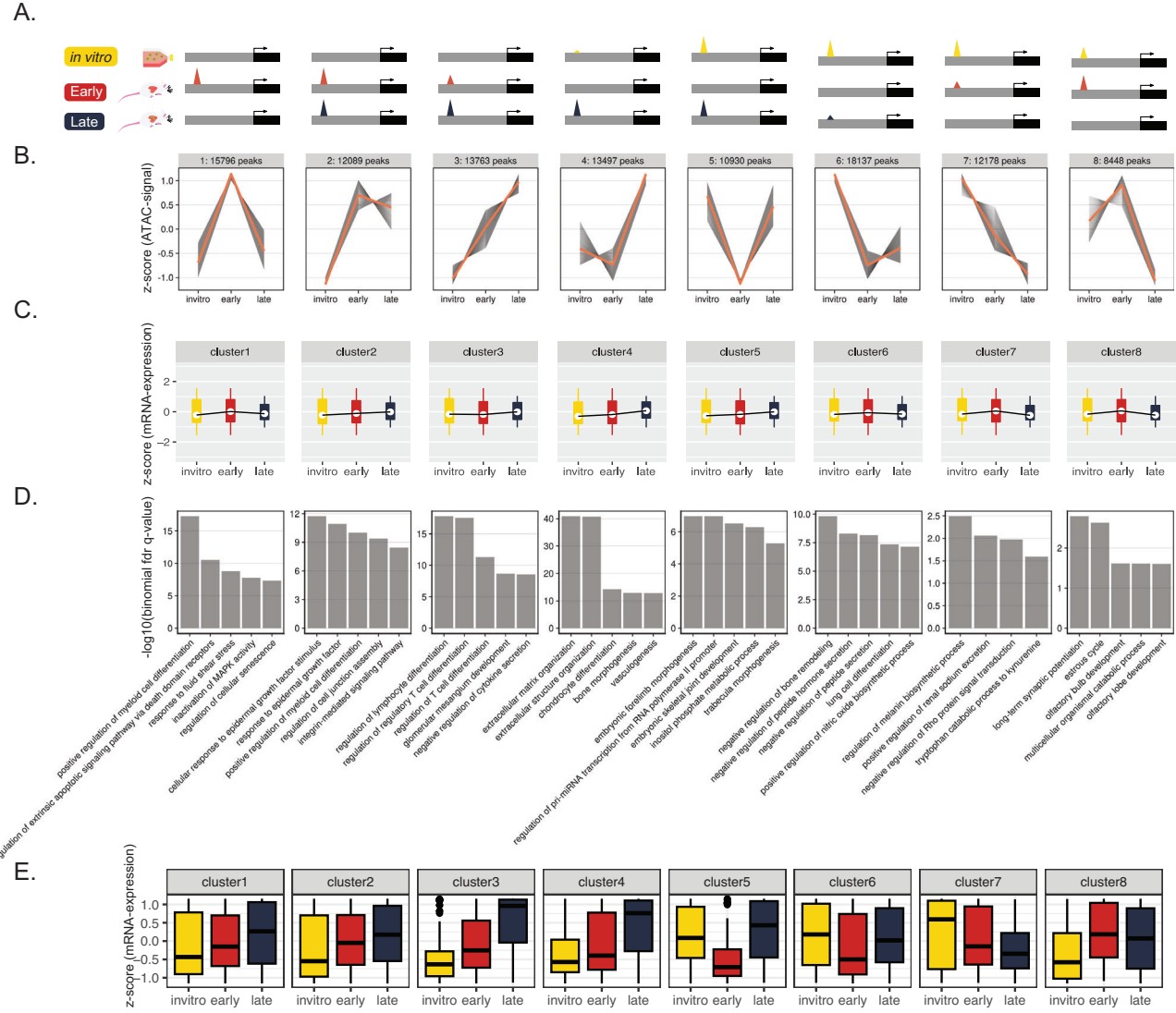

**Fig. 2 | Dynamic shifts in chromatin accessibility during osteosarcoma progression regulate temporally distinct metastatic programs. A** Diagram illustrating representative genome-browser views for peaks within each dynamic cluster. **B** K-means clustering partitions the "universe" of open chromatin based on their accessibility dynamics over time. Gray lines represent the dynamics of an individual peak, while the orange line represents the mean change for all peaks within a given cluster. **C** Z-scored mRNA expression for genes associated with peaks within each cluster. Boxplots represent the interquartile range where the top of the box is the third quartile, the bottom of the box is the first quartile, and the dot is the median. Whiskers extend to 1.5 times IQR. **D** Peak ontology based on GREAT for peaks within each cluster. Top significant terms are shown. **E** Z-scored mRNA expression for genes involved in the significant GREAT terms for each cluster. Cluster 1 *n* = 127, cluster 2 *n* = 146, cluster 3 *n* = 140, cluster 4 *n* = 359, cluster 5 *n* = 85, cluster 6 *n* = 91, cluster 7 *n* = 27, cluster 8 *n* = 53. Boxplots represent the interquartile range where the top of the box is the third quartile, the bottom of the box is the first quartile, and the midline is the median. Whiskers extend to 1.5 times IQR and dots represent outliers.

motifs for a variety of KLF factors. Interestingly, the late peaks were enriched for known mesenchymal factors like TWIST1 and SOX9/10, which have been previously shown to play a role in bone and cartilage development[21,22]. Of note, regions of the genome that lose accessibility early in metastasis (cluster 5) are highly enriched for CTCF motifs. Since CTCF is a structural factor involved in securing three-dimensional chromatin interactions, the loss of CTCF-enriched accessible sites may represent a loss of insulation and reorganization of chromatin topology[23]. To determine if these transcription factors were regulating the equivalent chromatin dynamics in other contexts, we performed the same analysis on the accessible chromatin clusters characterized in 143b-HOS-GFP (Supplementary Fig. 3A). NFkB family motifs were again enriched in the cluster 1 (early), while SOX9 and TWIST1 motifs were enriched for cluster 4 (late). In fact, the significantly enriched motifs for each

cluster were highly correlated between the two cell lines, demonstrating generalizability of our findings (Supplementary Fig. 3B).

To further narrow the list of putative regulators, we used the time course RNA-seq data to identify TFs whose expression dynamics mirrored the accessibility of the peaks within their respective cluster (Fig. 3B). Out of the factors enriched in the early-specific cluster, *NFKB2*, *NFKB1*, and *RELB* stood out as having an early-specific increase in expression, indicating a potential for differential activity at this time point. Excitingly, two of these genes, *NFKB2* and *RELB*, converge at the pathway level, as both are components of non-canonical NFkB signaling - a known driver of metastasis in other chromosomally unstable cancers[24]. For the putative pan in vivo cluster regulators, *KLF4*, *KLF12*, and *KLF13* all show an increase in expression at both in vivo time points compared to the same cells profiled in vitro. Lastly, we identified *SOX9*, *EBF1*, and *TWIST1* as likely

A.

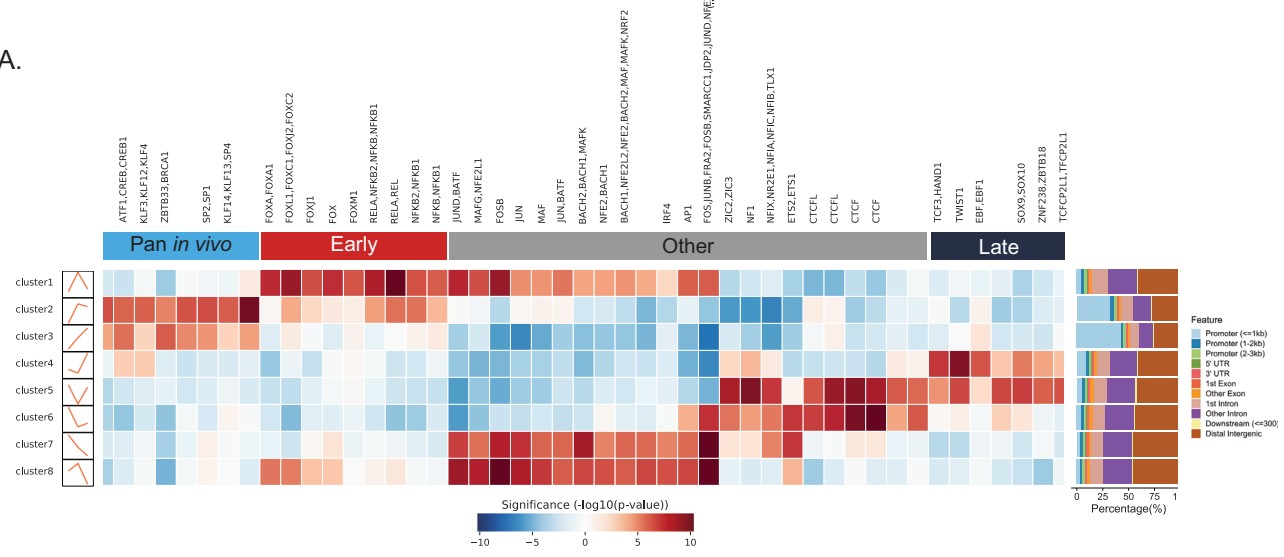

B.

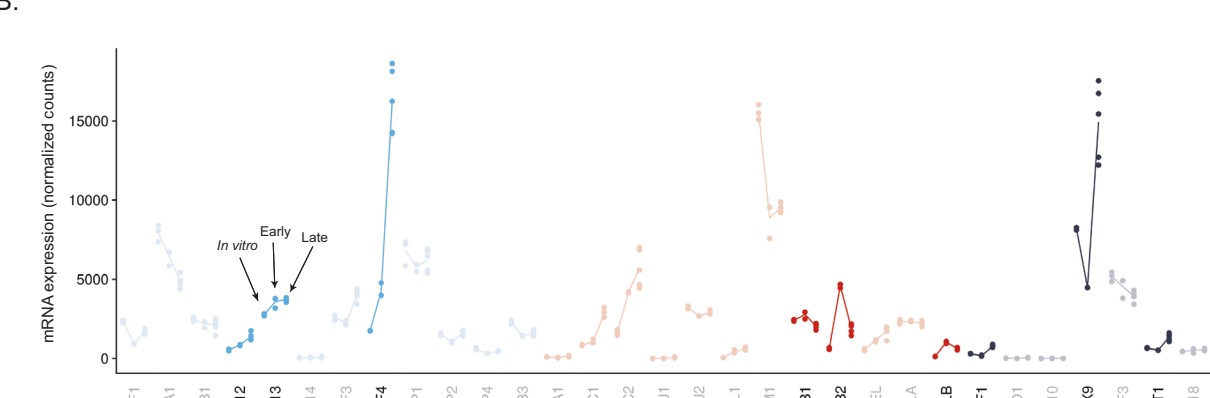

**Fig. 3 | Putative regulators of dynamic clusters are identified through motif-mining and gene expression changes. A** Transcription-factor motif enrichment heatmap displaying differentially enriched motifs for each dynamic cluster with genomic annotation of accessible regions contained within each cluster. Blank heatmap columns correspond to de novo motifs. **B** Gene expression dynamics of motif-enriched transcription factors during metastatic progression. Highlighted genes show expression changes that mirror the cluster's accessibility dynamics.

late-cluster regulators through this same approach. Many of these time point-specific patterns of TF gene expression were also observed in another metastatic osteosarcoma cell line (MNNG-HOS) when the cells were grown within lung explants (Supplementary Fig. 4). This supports the idea that a common set of regulators are responsible for the microenvironment-induced epigenomic reprogramming events across osteosarcomas.

While these accessibility dynamics could be due to active chromatin changes in vivo, they could also result from selection of subpopulations of cells that already possess the observed profiles in vitro (Supplementary Fig. 5A). We performed single-cell ATAC-seq on MG63.3-GFP cells grown in vitro to investigate these two possibilities (Supplementary Fig. 5B). Using the above TFs as markers for the temporal accessible chromatin landscapes, we projected either (1) their motif enrichment or (2) their promoter accessibility onto the scATAC-seq UMAP space (Supplementary Fig. 5C). This showed the unbiased clusters identified in vitro were not defined by temporal TF activity, indicating the dynamic peaks are likely a result of microenvironment-dependent reprogramming events, and not subclonal population shifts.

### The middle phase of lung colonization harbors condition-specific dependencies

While our data and integrated analyses define a sequence of epigenomic reprogramming events that occur during metastatic progression, it is unclear if these changes are necessary for metastasis to occur. We hypothesized if these changes were indeed driving metastasis, the upstream regulators would be essential. In vitro genome-scale CRISPR dropout screens have emerged in the past decade as a powerful way to assess gene dependency in a high-throughput manner. However, in vivo-specific biology is not captured with traditional workflows that solely use in vivo models to validate in vitro dependencies. In addition, screening coverage, and thus data quality, are limited by the number of cells that initially engraft within the lung. This makes genome-scale dropout screens in models of lung metastasis experimentally infeasible. To overcome these barriers, we designed a single guide RNA (sgRNA) library to target 78 TFs (along with 11 positive controls and 25 nontargeting negative control sgRNAs), and performed parallel experiments in vitro and in an in vivo model of lung metastasis (Fig. 4A, Supplementary Fig. 6). By combining 4 mice per replicate, we were able to

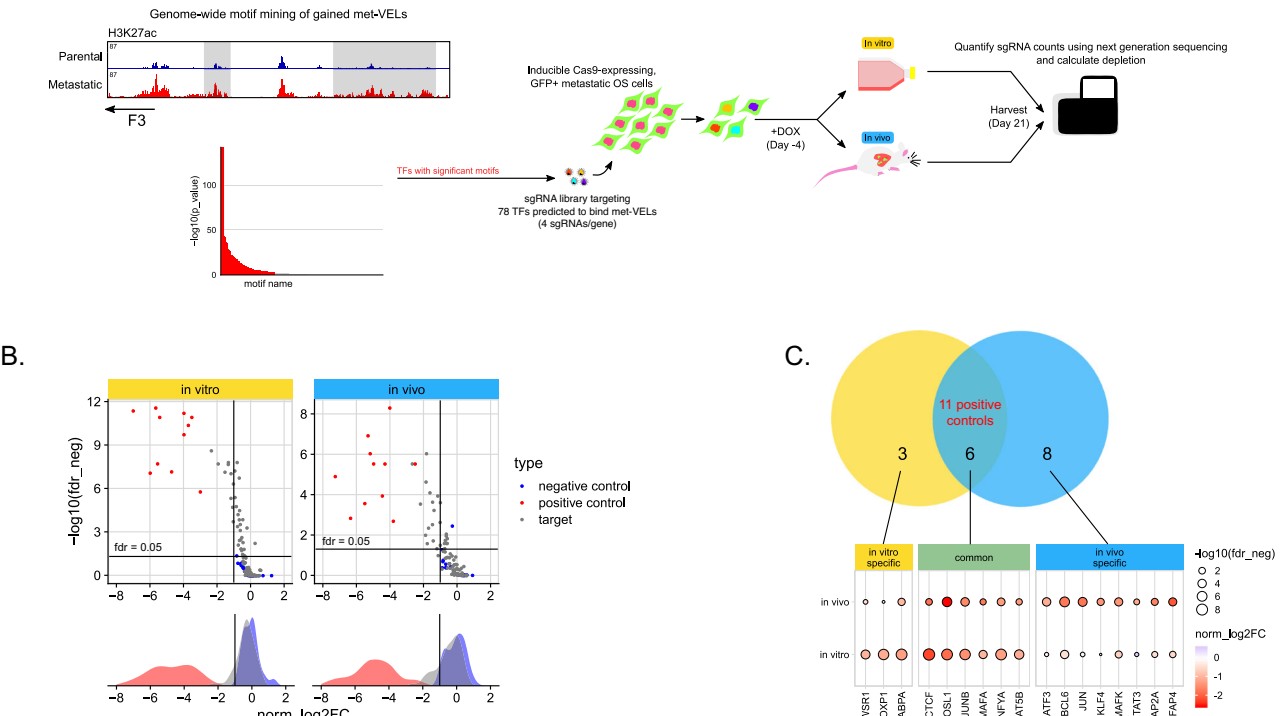

**Fig. 4 | Targeted CRISPR screen reveals condition-specific transcription-factor dependencies in metastatic osteosarcoma. A** Experimental workflow for in vitro and in vivo CRISPR screens in a metastatic osteosarcoma cell line. **B** One-sided volcano plots and marginal density plots displaying distribution of gene targets included in the screens. Gene-level statistics are shown. **C** Set of genes called as hits in each screen. The cutoff used for calling hits was an fdr neg <0.05 and log(fold-change) < -1.

achieve a screening coverage of 400× in vivo. Both screens were performed using the MG63.3-GFP-Cas9i cell line previously described[13].

We confirmed significant depletion of sgRNAs targeting all 11 positive controls in both arms of the screen. In addition, the non-target negative controls did not affect cell fitness, collectively demonstrating the quality of both screens (Fig. 4B). In total, the knockout of 17 TFs were shown to reduce cell growth both in vitro and within the lung microenvironment. Excitingly, 8 TFs were classified as metastasis-specific dependency genes, with fewer classified as in vitro-specific (3 genes) or common dependencies (6 genes other than the positive controls) (Fig. 4C). The 8 metastasis-specific dependency genes included multiple TFs from the AP−1 factor family, which has previously been implicated in osteosarcoma metastasis[7,25]. Besides these, we found *KLF4*, *STAT3*, *BCL6*, *TFAP2A*, and *TFAP4* as potential drivers of osteosarcoma metastasis. To further investigate the context-specificity of our genes of interest, we probed publicly available whole genome CRISPR screen data in 990 cell lines from the Dependency Map (DepMap) database[26,27]. While *TFAP4* was an essential gene in 48% of the cell lines and *BCL6* was a lymphoma-specific essential gene, the other 6 in vivo hits showed little evidence of depletion across all cancer types, including 9 osteosarcoma cell lines (Supplementary Fig. 7). This finding further highlights the importance of screening directly in the metastatic microenvironment, and that our in vivo hits are selectively important for lung metastasis, and not other osteosarcoma growth contexts.

### Genetic and pharmacological inhibition of pro-metastatic transcription factors prevents lung metastasis

We reasoned that if the TF-hits in the CRISPR screen were genuine metastasis-specific dependencies, then inhibiting them should impair lung metastasis but spare in vitro growth. We first sought to show that these factors demonstrated context-specific importance outside of the screening setting. Since *KLF4* was not only an in vivo-specific hit from our screen, but also a likely mediator of the in vivo-specific chromatin changes observed, we chose to start with this factor. Using the top sgRNA from our screening library, we knocked out *KLF4* in a pooled format in MG63.3-GFP-Cas9i (Fig. 5A). As a control, we transduced the same cell line with a nontargeting sgRNA. We observed that knocking out *KLF4* did not decrease the viability of metastatic osteosarcoma cells grown over the course of 7 days in vitro (Fig. 5B). In line with these findings, knocking out *KLF4* also had no impact on the in vitro enhancer landscape of MG63.3-GFP cells ($r = 0.98$) (Supplementary Fig. 8A, B). Using CRISPResso2, we determined the majority (85%) of ChIP-seq reads mapped to the *KLF4* locus in our knockout cells contained edits predicted to disrupt KLF4 expression (Supplementary Fig. 8C)[28]. This was not the case for the *KLF4* wild-type ChIP-seq. This confirmed the similarity of the enhancer landscapes between the two cell lines was not simply due to outgrowth of *KLF4* wild-type cells within our pool.

In order to study the effect of *KLF4* knockout on metastatic capability, we used an ex vivo <u>Pu</u>lmonary <u>M</u>etastasis <u>A</u>ssay (PuMA)− secondary model of lung metastasis[13,29]. In this system, metastatic osteosarcoma cells are injected into the lungs of mice through the tail vein (Fig. 5C). The mice are immediately euthanized, after which lung slices are grown at an air liquid interface ex vivo. By monitoring the tumor burden by GFP-positive area within each lung slice, we saw that knocking out *KLF4* decreased the ability of the cells to grow within the context of the metastatic microenvironment. At every time point imaged, the level of tumor burden was significantly lower in the *KLF4* knockout cell line than the non-target control line (Fig. 5D, E). While tumor burden did increase over time for both

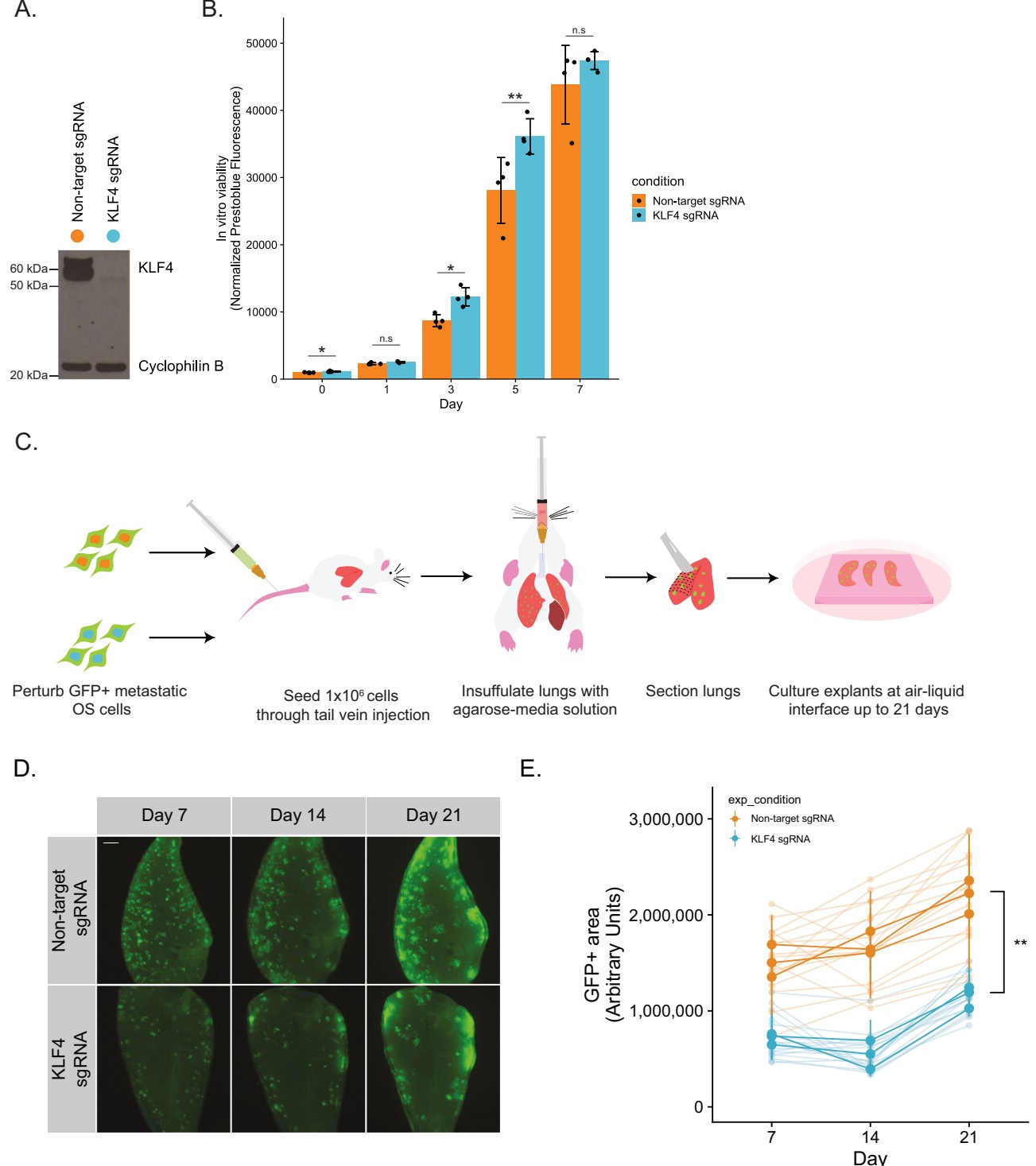

**Fig. 5 | Kruppel-like factor 4 (KLF4) is a bona fide metastasis-specific dependency. A** Western blot showing KLF4 knockout in a metastatic osteosarcoma cell line compared to non-target control transduced cells. **B** In vitro cell viability assay using PrestoBlue reagent. Data are represented as mean ± SD, with $n = 4$ wells per cell line for each time point. *** <0.0005, ** <0.005, * <0.05, n.s > 0.05. **C** Illustration of the workflow for the ex vivo pulmonary metastasis assay (PuMA). **D** Longitudinal imaging of representative lung sections for each cell line. Scale bar denotes 500 μm. **E** Quantification of metastatic burden in the PuMA assay using GFP+ area. Data are represented as mean ± SD, with $n = 3$ individual mice assessed per condition with $n = 8$ lung sections quantified per mouse. ** <0.005.

conditions, this could be representative of cells within the pool of *KLF4* knockout cells that escaped editing of the *KLF4* locus. However, we cannot rule out the possibility that KLF4 is only partially essential for lung metastasis, or that metastasizing cells can circumvent KLF4 dependence through usage of alternative transcriptional machinery.

Despite the clear importance of KLF4 in promoting metastasis in our models, therapeutic inhibitors of this factor have not yet been developed. However, other in vivo dependencies from our screen, such as STAT3, have chemical probes that can be used to impair their function. We thus sought to determine if a STAT3 inhibitor could selectively target osteosarcoma lung metastasis.

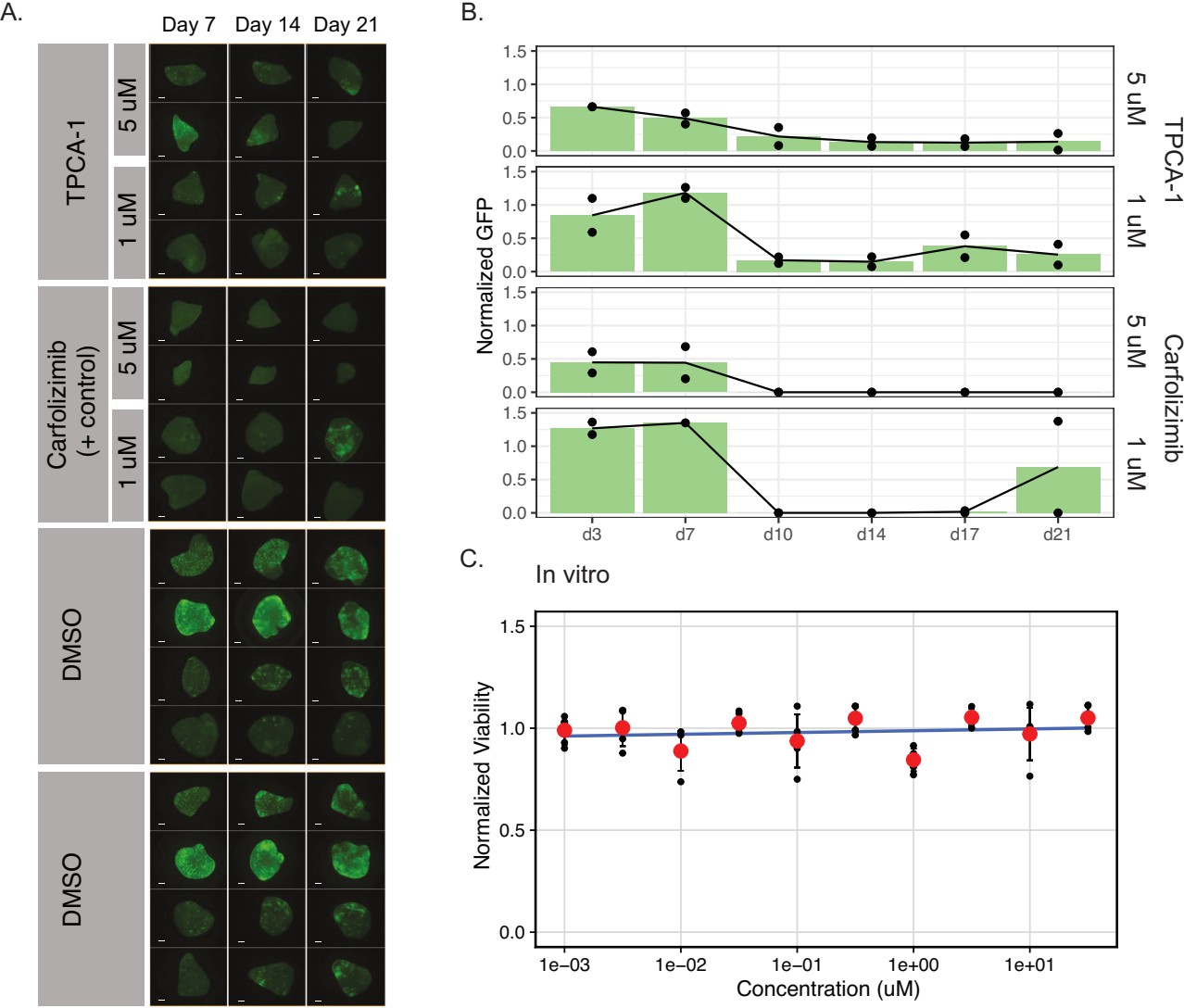

**Fig. 6 | STAT3 inhibition with TPCA-1 attenuates metastasis without in vitro cell viability effects. A** Longitudinal imaging of PuMA lung sections treated with TPCA-1, Carfolizimib (positive control), or DMSO (negative control). Scale bar denotes 500 μm. **B** Quantification of metastatic burden for each condition based on GFP signal. Treatment conditions were normalized to DMSO control sections from the same mouse to account for intermouse variability. **C** In vitro dose response curve for MG63.3 cells treated with TPCA-1. PrestoBlue reagent was used to measure cell viability. Data are displayed as mean ± SD, with $n = 5$ wells per concentration.

Using the PuMA system, we found that the STAT3 inhibitor TPCA−1 prevented ex vivo lung metastasis at 5 μM and 1 μM over the course of 21 days (Fig. 6A, B). Confirming our genetic findings that *STAT3* is an in vivo-specific dependency, the same cells were unaffected by TPCA-1 when grown outside of the lung microenvironment (Fig. 6C). These data demonstrate that our approach identified druggable metastasis dependency genes that may serve as therapeutic targets for treating metastatic osteosarcoma.

**Metastasis dependency transcription factors represent a transcriptional addiction**

While some of the pro-metastasis TFs are targetable with specific inhibitors, not all TFs are able to be directly targeted in this manner. However, many of the metastasis-specific hits from our in vivo screen are upregulated at the mRNA level within the lung micro-environment in 2 cell lines (Fig. 7A, B). This indicates that meta-static osteosarcoma cells may depend on the increased expression of these TFs for successful lung colonization. In other words, the dependence on this specific set of transcription factors may represent a metastasis-specific transcriptional addiction. We

wondered whether this increase in expression of pro-metastasis TFs could be a potential vulnerability for metastatic osteo-sarcoma cells.

Experimental probes that target the transcriptional machinery such as JQ1 and THZ1 have been used to determine transcriptional addictions in cancer cells[30–32]. Treatment of the metastatic osteo-sarcoma cell line MNNG-HOS with JQ1 has been shown to block tumor growth within the lung microenvironment[7]. To determine if the anti-metastasis property of JQ1 correlated with downregulation of any of the pro-metastatic transcription factors identified, we reanalyzed data profiling the transcriptome of MNNG-HOS grown in the PuMA system with and without drug. Strikingly, 6 of the 8 in vivo hits from MG63.3 showed an increase in expression when MNNG-HOS cells were grown within the context of the lung microenvironment (Fig. 7B). The increase in expression of all 6 of these factors was attenuated by exposure to JQ1 (Fig. 7C, D). Of these 6 factors, *KLF4, STAT3,* and *JUN* were further upregulated in metastases in an inde-pendent cohort of osteosarcoma samples from St. Jude (Fig. 7E). Altogether, these data indicate that transcription-factor dependen-cies identified in one cell line could be more broadly relevant to

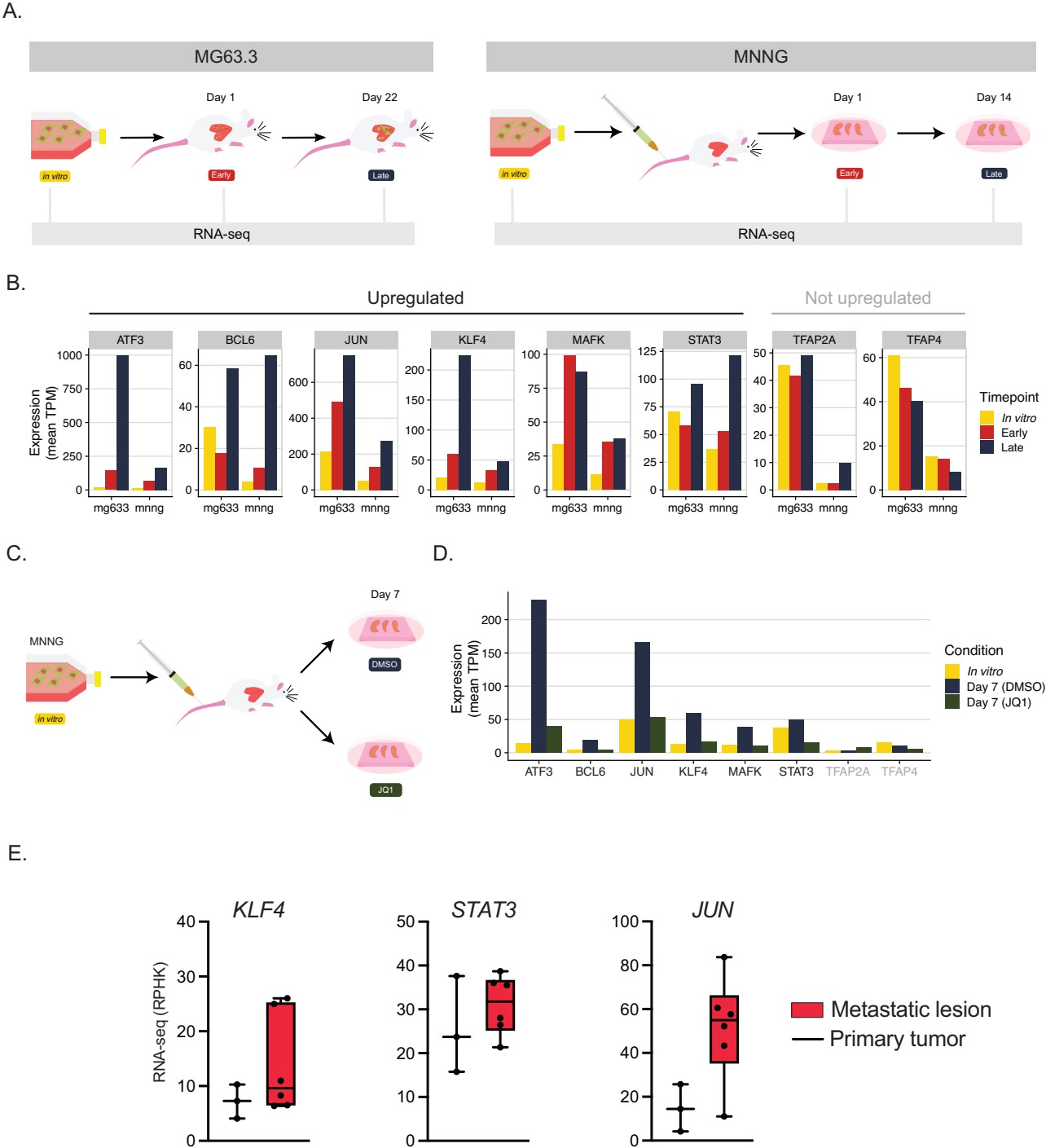

**Fig. 7 | Transcriptional inhibition with JQ1 prevents lung metastasis and abrogates the increased expression of pro-metastatic transcription factors within the lung microenvironment. A** Schematic comparing RNA-seq profiling experiments of MG63.3 and MNNG cell lines. MG63.3 early and late conditions were isolated from a fully in vivo model of metastasis, whereas MNNG cells were isolated from the ex vivo PuMA model. **B** Barplots demonstrating RNA expression dynamics of in vivo CRISPR screen hits in both MG63.3 and MNNG cells. **C** Schematic illustration of experiment assessing effect of JQ1 on MNNG gene expression during lung metastasis. **D** RNA expression of in vivo CRISPR screen hits in the MNNG cell line, with and without JQ1 treatment. **E** RNA expression of 3 in vivo CRISPR screen hits upregulated in patient samples of metastatic lesions when compared to primary tumors. Primary tumor *n* = 3, metastatic lesion *n* = 5. Boxplots represent the interquartile range where the top of the box is third quartile, the bottom of the box is first quartile, and the midline is the median. Whiskers extend to 1.5 times IQR.

human osteosarcoma, and transcriptional inhibition may serve as a general therapeutic strategy.

## Discussion

Despite metastasis being a devastating clinical problem, our understanding of the epigenetic mechanisms associated with the process have been limited to the biological bookends. While mutational drivers of cancer processes are preserved during tumor progression, the plasticity of the epigenome means epigenetic drivers can be transient. This can make them invisible with traditional end point comparisons. The past focus on comparing primary tumors to late-stage metastatic disease has limited our understanding of the complex phase of lung colonization,

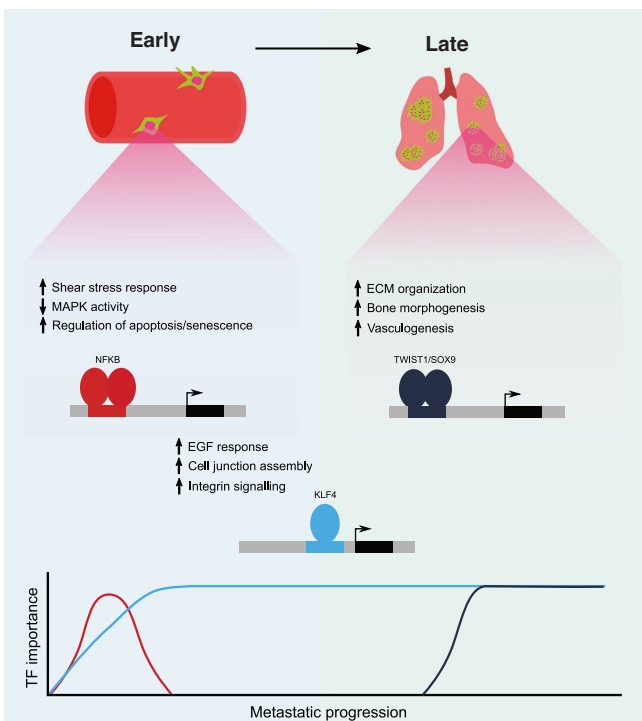

**Fig. 8 | Model for transcription-factor driven lung metastasis in osteosarcoma.** Schematic model illustrating the role of dynamic modulation of the epigenome during metastatic colonization in osteosarcoma.

which remains a black box. Here we show this critical period of osteosarcoma metastasis is composed of a continuum of changes that occur as cancer cells interact with their new microenvironment.

Waddington's epigenetic landscape for embryonic development depicts a series of ridges and valleys that represent paths a pluripotent cell can take to differentiate into a variety of somatic cell types. Along each of these paths, the differentiating cell encounters branch points that end in different tissues, with the apex of this branch point representing intermediate cell states. In a similar way, we envision a metastasizing cell as rolling down an analogous epigenetic landscape. Where fully differentiated normal tissue is present at the bottom of Waddington's landscape, our model places metastatic lesions indicative of end-stage disease. Our work defines the epigenetic path an osteosarcoma cell takes to colonize the lung, identifying multiple intermediate cell states along the way.

Importantly, we found these specific epigenetic reprogramming events present vulnerabilities that can be targeted at multiple distinct levels of gene regulation. Upstream transcription factors driving this reprogramming can play temporally specific roles in the process of lung colonization. However other factors like the master transcription-factor KLF4 are ubiquitously important. We show through an in vivo functional genomics approach and subsequent validation experiments that KLF4 is indeed required for successful outgrowth in a murine pulmonary metastasis xenograft model. This same factor was totally dispensable for the same cells to grow in vitro, emphasizing its context-specific importance. More work is required to determine correlates of KLF4 dependence, such as patient sex, tumor location, or specific genomic/epigenomic alterations (i.e. amplification of *MYC*, mutations in *TP53*, specific variant enhancers, etc.).

In addition to KLF4, we identify seven other transcription factors that are essential for lung metastasis, but unimportant for in vitro proliferation. One of these factors, STAT3, can be directly targeted in an experimental setting using the drug TPCA-1. However, more broadly, the general addiction of metastasizing cancer cells to the

expression of TF dependency genes is indirectly targetable through transcriptional inhibition with JQ1. This provides further evidence that multiple different approaches to targeting the transcriptional axis show promise as therapeutic strategies in metastatic osteosarcoma.

Although we focus mostly on pan in vivo dependencies, another exciting finding from our work is that the different stages of metastasis may be targeted by inhibiting the specific transcriptional machinery required for each phase of lung colonization (Fig. 8). This indicates targeting the various clinical manifestations of the disease will likely require different drugs. For example, targeting early regulators will likely be optimal for preventing early phases of colonization, whereas late regulators may be better targets for more established metastatic disease.

Our work also provides insights into the basic biology underlying osteosarcoma lung metastasis. Both SOX9 and KLF4 have been shown to play key roles in normal bone development. KLF4 inhibits osteoblast differentiation to regulate bone homeostasis, while SOX9 serves as a master regulator of chondrocytes - an intermediate cell type during osteoblastogenesis[33,34]. Thus, our data suggest metastatic competence may depend on osteosarcoma's ability to transiently dedifferentiate from an osteoblast-like state. This is in line with findings in prostate cancer showing early developmental programs are important for metastasis[35].

In totality, we develop a strategy combining metastasis time point experiments and in vivo functional genomics to elaborate on seminal work describing the importance of the epigenome to the metastatic phenotype across multiple cancers[7–9,36]. While other studies focused on end point comparisons, we are the first to show distinct changes at the level of chromatin occur multiple times throughout lung colonization. Furthermore, these temporally resolved changes point in the direction of a host of metastasis dependency genes that lie throughout the dynamic epigenomic landscape of metastasizing cells. Experimentally, this study demonstrates the importance of using in vivo models of cancer processes in the earliest stages of discovery research. Translationally, we believe the gap in knowledge this study fills is a critical step towards improving treatments for osteosarcoma patients. Many osteosarcoma patients present with undetectable micrometastases, limiting the utility of therapies targeting dissemination and seeding from the primary tumor, as this stage of disease has passed and is largely irrelevant to the patient's outcome. Established macrometastases mean the cancer cells already have a stronghold in the patient's lung and through accelerated growth are ready to evolve resistance to therapies. In both scenarios we have likely missed a key therapeutic window of opportunity. Defining intermediate epigenetic states and upstream regulators required for cells to successfully colonize the secondary organ allows us to target metastasis when it is at its most vulnerable. Since patients are likely to possess metastatic cells at different stages of progression simultaneously, a combination therapy tailored to address each distinct biology may be optimal. Although our work identifies reproducible epigenetic changes during osteosarcoma lung colonization, additional studies are needed to assess if the same regulators play roles in other lung-metastatic cancers, or even other cancers of the same lineage. This same approach could also be used to probe the dynamic nature of other aspects of cancer progression, such as primary tumor formation or colonization of other distal organs.

### Limitations of study

First, our study is focused primarily on two different osteosarcoma models (MG63.3 and 143b). Despite commonality between these well-established and genetically distinct cell lines, we know osteosarcoma is a heterogeneous cancer. Similar experiments in other models that represent the larger spectrum of osteosarcomas would reinforce the broad applicability of these findings. Second,

although we identify intermediate transcriptional states that occur during lung colonization, our study is limited to three time points and thus represents only a snapshot of the full epigenetic trajectory of lung colonization. Higher-resolution temporal profiling could help pinpoint additional dependencies. Third, the current study does not consider the contribution of the immune system to the metastatic colonization process, given our experimental system whereby human cells are seeded to the lungs of immunocompromised mice. Analogous approaches using syngeneic models could help in this regard. Lastly, the epigenetic states characterized in our study reflect responses to numerous stresses the osteosarcoma cells encounter in the lung microenvironment. Future studies profiling the epigenome of osteosarcoma cells under specific stresses (shear stress, hypoxia, co-culture, anchorage independent growth, etc.) could further refine the basis of the temporal chromatin changes observed here.

## Methods

### Ethics statement
All animal experiments were performed in accordance with the Case Western Reserve University (CWRU) Institutional Animal Care and Use Committee's guidelines.

### Cell culture
The human osteosarcoma cell lines MG63.3-GFP, MG63.3-GFP-Cas9i, and 143b-HOS-GFP were obtained, generated, and cultured as previously described[13]. The MNNG-HOS data was a reanalysis of a public dataset. Therefore, methods regarding this cell line are the same as previously described[7]. Routine testing to ensure the absence of mycoplasma was performed using a custom PCR-based assay. Unique cell lines described in this paper are available from the authors upon request.

### Mouse studies
All mouse experiments were performed using NOD scid gamma mice purchased from the CWRU Athymic Animal & Preclinical Therapeutics core facility. Mice were housed in ultraclean facilities in accordance with protocols approved by the CWRU Institutional Animal Care and Use Committee. Experiments were designed to minimize mouse use, while optimizing statistical power, based on our extensive experience with xenograft models of metastasis. Mice were standardized for sex and age for each individual experiment. Since no subjective measurements were used for any of the described experiments, researchers were not blinded to group assignments.

### Ex vivo lung metastasis assay
The ex vivo pulmonary metastasis assay was performed as previously described[7,29]. In brief, $1 \times 10^6$ cells were injected into 10–12 week old female NOD scid gamma mice through the lateral tail vein. Mice were immediately sacrificed post-injection. The lungs of the mice were insufflated with a 50% medium/agarose solution. The lungs were then resected and placed in ice cold 1× PBS for 30 min to allow the medium/agarose solution to polymerize. After 30 min, individual lobes of the lung were cut into transverse sections and placed on Gelfoam (Pfizer, catalog 00300090315085) that had been equilibrated in medium in 6-well plates for 24 h at 37 °C. Sections were then cultured ex vivo for 21 days at 37 °C and 5% CO$_2$. Each section was removed from the Gelfoam and imaged in a new cell culture plate at the time points listed in each experiment to assess relative tumor burden by GFP+ area. Fluorescent imaging was performed on the Operetta High Content Imaging System (PerkinElmer), and quantification of tumor burden was done via the Acapella Image Analysis software (PerkinElmer). For the KLF4 knockout experiment, three mice were used per condition, with 6 lung sections cultured per mouse.

### In vitro cell viability assay
To measure the effect of *KLF4* knockout on cell viability in vitro, 500 *KLF4* knockout cells or non-target control cells were plated in 96-well plates in 90 μL of medium, with six wells seeded per condition. In total, five 96-well plates were seeded to allow measuring of viability at days 0, 1, 3, 5, and 7. After seeding, cells were allowed to adhere overnight before measuring viability at day 0 using PrestoBlue reagent (ThermoFisher, A13261). To do this, 10 μL of PrestoBlue was added to each well and allowed to incubate at 37 °C for 1 h. Fluorescence was then measured on a Synergy Neo2 plate reader (BioTek). For cells cultured past day 0, medium with 1 μg/mL doxycycline was added on day 0. Viability was measured at each time point as described above.

### Western blot
Cells were lysed at 4 °C in RIPA buffer supplemented with protease inhibitors (Roche, 4693159001). Protein concentrations were measured using a BCA Assay Kit (Thermo Fisher, 23225), and 30 μg of total protein was resolved on precast 4–12% Bis-Tris gels (Invitrogen, NP0321BOX) and transferred to PVDF membranes (Bio-Rad, 1704157) using the trans-blot turbo transfer system (Bio-Rad, 1704150). The membranes were blocked with 5% dry milk in PBS supplemented with 0.2% Tween-20 (PBS-T) at room temperature for 1 h and then incubated overnight at 4 °C with the following primary antibodies: KLF4 at 0.5 μg/mL (R&D, AF3640) and cyclophilin B at 1:10,000 dilution (abcam, catalog ab16045). Chemiluminescent detection was performed with HRP-conjugated secondary antibodies purchased from Thermo Scientific at 1:10,000 dilution (anti-goat, RA2143996; anti-rabbit, 31460) and developed using Genemate Blue Ultra-Autoradiography film (VWR, 490001-930).

### CRISPR screen and individual inducible knockout cell line generation
The custom CRISPR-screen library was generated as previously described[37]. Three hundred and sixty sgRNAs targeting the TF targets were pulled from the genome-wide Brunello library[38]. In addition, 25 sgRNAs without recognition sites in the human genome were included as negative controls. All sgRNAs were synthesized (CustomArray) and cloned into pLV-U6-gRNA-UbC-DsRed-P2A-Bsr; a gift from Charles Gersbach (Addgene, plasmid 83919). Lentivirus was produced with LentiX Packaging Single Shots (Clontech, 631278) according to the manufacturer's protocol and was used to transduce MG63.3-GFP-Cas9i cells at an MOI of -0.3. After selection of the successfully transduced cells with 5 μg/mL blasticidin, screening pools were expanded and used for the in vitro and in vivo screens. All single knockout cell lines were generated as described above, with the exception that single guides were cloned into pLV-U6-gRNA-UbC-DsRed-P2A-Bsr according to the Broad institute protocol. Knockout induction was performed in vitro by exposing cells to 1 μg/mL doxycycline hyclate (Cayman, 14422).

### sgRNA sequences
sgRNA sequences were as follows: *KLF4*, forward, KO: CACCGGAGCGATACTCACGTTATTCG; *KLF4*, reverse, KO: AAACCGAATAACGTGAGTATCGCTC; nontargeting control-1, forward, KO: CACCGAAAAAGCTTCCGCCTGATGG; and nontargeting control-1, reverse, KO: AAACCCATCAGGCGGAAGCTTTTTC.

### In vitro CRISPR dropout screen
For the in vitro screen, $5 \times 10^5$ cells were seeded in triplicate in T75 flasks. Cells were maintained for 21 days in the presence of 1 μg/mL doxycycline hyclate, with $5 \times 10^5$ cells reseeded at each passage to maintain 500× library coverage. At the end of the 21 days, $5 \times 10^5$ cells were collected for genomic DNA extraction using red blood cell lysis solution (Qiagen, 28606).

## In vivo CRISPR dropout screen

Fifteen mice were placed on water with 2 mg/mL doxycycline hyclate for 4 days prior to the beginning of the experiment. In parallel, $1 \times 10^6$ cells were seeded in triplicate and cultured in the presence of 1 ug/mL doxycycline hyclate to induce Cas9 expression. Cells were then expanded in vitro for 4 days prior to injection. Three groups of 5 mice (15 total) were seeded with $1 \times 10^6$ cells each. Mice were maintained on 2 mg/mL doxycycline for the duration of the experiment. After 21 days of in vivo growth, cells were isolated from the lungs of individual mice. In brief, lungs were removed from post-mortem mice and dissociated using the Miltenyi Human Tumor Dissociation kit (Miltenyi, 130-095-929) and GentleMACS dissociator (Miltenyi, 130-093-235) according to the manufacturer's protocol for medium tumors. To enhance the purity of human cells within the final cell suspension, mouse cells were removed through negative selection using the Miltenyi Mouse Cell Depletion Kit (Miltenyi, 130-104-694). Genomic DNA was then extracted from the purified population of human cells with red blood cell lysis solution and used to prepare sequencing libraries as previously described[37]. In brief, 2000 μg of genomic DNA was used as a template for PCR, split across 8 separate reactions for each sample. Phusion high-fidelity master mix was used for all PCR reactions. Libraries were then purified using PCR Clean DX beads (Aline Biosciences), pooled, and paired-end sequenced on a MiSeq (Illumina) using custom read and index primers. Although each group of 5 mice was analyzed as a single replicate to obtain a screening coverage of 500×, individual libraries were generated for each mouse.

Analysis was performed using the web-based CRISPRCloud2[39]. Dropout for both the in vivo and in vitro screens were calculated compared to an input day 0 time point. For the in vivo screen, all 5 mice from each of the three groups were assigned as a single replicate. Hits for each screen were called based on a false-discovery rate <0.05, and a log2(fold-change) < −1.

## Single-cell ATAC-seq

MG63.3 cells were isolated and processed according to the Nuclei Isolation for Single Cell ATAC Sequencing Demonstrated Protocol (10X Genomics, CG000169) with the following modifications: 1% BSA in PBS was used for the washes, and cells were lysed for 3 min on ice. Sequencing libraries were created using the Chromium Single Cell ATAC Reagent Kits User Guide (10X Genomic, CG000168). Briefly, 5000 nuclei were targeted and tagmented in bulk. Nuclei were then portioned into Gel Beads-in-emulsion (GEMs) with a unique cell barcode per single nucleus. GEMs were amplified first in a linear amplification PCR, after which GEMs were broken and PCR was used to add a sample index and Illumina sequencing handles (P5/P7). Libraries were sequenced at the University of Chicago Genomics Facility on an Illumina HiSeq 4000 with paired-end 100 bp reads.

Single-cell ATAC-seq data were aligned and processed using Cell Ranger ATAC v1.2 with the hg19 reference genome. The peak cell matrix and fragment file were further processed using Seurat v4.0.1 to create a Seurat object of the class ChromatinAssay[40]. Using Seurat v4.0.1, latent semantic indexing (LSI) was used to perform normalization, feature selection, and linear dimensional reduction. UMAP reduction was performed using LSI on dimensions 2 through 30 and cells were plotted in UMAP space.

To assess promoter accessibility of marker TFs for each cluster at the single-cell level, Seurat's FeaturePlot function was used. The promoter for each gene was defined as the genomic region 1 kb upstream and downstream from the transcription start site obtained from the UCSC Genome Browser. For visualization, the accessibility score at the promoter of each gene was projected onto the single-cell UMAP space.

To determine the distribution of motif enrichment for each marker TF, chromvar was used. First, the motif position frequency matrix was created using motif information from the JASPAR database (JASPAR2020 matrix for species 9606, homo sapiens). Next, chromvar was used to calculate a motif score per cell for each TF. Seurat's FeaturePlot function was then used to visualize the motif enrichment score per cell for each of the cluster defining transcription factors in the UMAP space.

## ATAC-seq

ATAC-seq was performed using the omni-ATAC-seq protocol previously described[41]. For cells profiled in vitro, $2.5 \times 10^4$, $5 \times 10^4$, or $1 \times 10^5$ cells were used. For the in vivo conditions, $1 \times 10^5$ cells were used. Prior to preparation of the in vivo libraries, human osteosarcoma cells were isolated from mice 1 day or 22 days post-tail vein injection. In brief, lungs were surgically removed from post-mortem mice and dissociated using the Miltenyi Human Tumor Dissociation kit (Miltenyi, 130-095-929) and GentleMACS dissociator (Miltenyi, 130-093-235) according to the manufacturer's protocol for medium tumors. To enhance the purity of human cells within the final cell suspension, mouse cells were removed through negative selection using the Miltenyi Mouse Cell Depletion Kit (Miltenyi, 130-104-694). For the early condition, the final cell suspensions from 4 mice were combined prior to performing ATAC-seq for each replicate (3 replicates, 12 mice). For the late condition, each replicate was generated from an individual mouse (5 replicates, 5 mice). MG63.3 libraries were sequenced at the CWRU Genomics Core Facility on an Illumina NextSeq (high-output flowcell) with paired-end 75 bp reads. 143b-HOS-GFP libraries were sequenced at MedGenome with paired-end 100 bp reads.

Reads were aligned to human genome reference hg19 with BWA-MEM[42]. Although the purity of human cells was enriched experimentally through magnetic sorting, a residual population of mouse cells were still present in the purified population. To discount these cells from downstream analysis, we also aligned each sample to mouse genome reference mm9 and used the R package XenofilteR to selectively remove reads with better alignment to the mouse genome than human[43]. Duplicate reads were removed from the filtered BAM files. The de-duplicated BAMs were then used to call peaks with Genrich (https://github.com/jsh58/Genrich) on ATAC-seq mode with default settings. Bigwig tracks were generated using deeptools "bamCoverage" with normalization by RPGC.

To partition the landscape of open chromatin identified based on accessibility dynamics, a z-scored matrix was created from the quantile-normalized fpkm values across the universe of ATAC-seq peaks across all time points. Peaks with a coefficient of variation <10% were binned into a pseudo "static" cluster. The remaining dynamic regions were then clustered with k-means clustering. Bedtools2 was used for all genome arithmetic during the analysis process.

## RNA-seq

Cells grown in vitro and cells remaining after the in vivo ATAC-seq were lysed using TRIzol (Invitrogen, 15596026). RNA was subsequently extracted by transferring the aqueous phase from the TRIzol-chloroform extraction to RNeasy columns (Invitrogen, 74104). The rest of the RNA extraction was performed according to the RNeasy manufacturer protocol. Purified RNA was sent to MedGenome for library preparation and sequencing. MG63.3 libraries were prepared using the Takara SMARTer Stranded Total RNA-Seq Kit v2 - Pico Input Mammalian (Takara, 634411) while 143b-HOS-GFP libraries were prepared using the SMART-Seq v4 Ultra Low Input RNA Kit (Takara). All libraries were sequenced paired-end 100/150 bp.

Raw RNA-seq reads were aligned to hg19 and mm9 using HISAT2, and XenofilteR was again used to remove reads that aligned better to the mouse genome than the human genome[43,44]. Transcripts per million were calculated for each gene across all time points.

For the patient analysis, bulk RNA-seq data for paired primary and metastatic tumors were obtained from the St. Jude Children's Research

Hospital Childhood Solid Tumor Network (CSTN)[45]. Clinical details from the samples used are available in Supplementary Table 1. Transcript per million reads (TPM) of each gene was measured using RSEM version 1.3.3 that calculated TPM by reassigning multiple alignments of STAR version 2.5.3a to target genes via a maximum likelihood estimation framework. Expected counts of coding genes from RSEM were totaled by sample to yield a sum of all expected counts. The expected counts for each gene were then divided by the total sum of expected counts and multiplied by 100,000. The resulting gene expression values were then plotted in GraphPad PRISM, binned into groups depending on metastatic status of the tumors.

### ChIP-seq
ChIP-seq was performed as previously described using an antibody targeting H3K27ac (Abcam, ab4729)[46]. In brief, 5 million MG63.3 sgKLF4 or sgNT cells were fixed with methanol free formaldehyde for 10 minutes. Nuclei were extracted and chromatin was sheared for 7 min using the Covaris S2 AFA focused ultra sonicator (Duty factor 5%, intensity 4, 200 cycles/burst). Libraries were prepared as previously described[46,47]. Libraries were sequenced with paired-end, 150 bp reads.

### ChIP-Seq data processing
Cutadapt v1.9.1 was used to remove paired-end adapter sequences and discard reads with a length less than 20 bp[48]. FASTQs were aligned to hg19 using BWA-MEM with default parameters in paired-end mode. Output SAM files were converted to binary (BAM) format, sorted, indexed, and PCR duplicates were removed using SAMtools v1.10[49]. Peaks were detected with MACS v2.1.2 with the --broad flag set[50]. DeepTools v3.2.0 was used to generate RPGC-normalized bigWig tracks with 50 bp bin sizes from the final sample BAM files[51]. BigWigs were visualized on the Integrative Genomics Viewer in order to assess pronounced track irregularities or low signal-to-noise ratio[52]. Bedtools2 was used for all genome arithmetic involving ChIP-seq data. CRISPResso2 was used to confirm editing of KLF4 within the sgKLF4 H3K27ac ChIP-seq data[28].

### GREAT analysis
Bed files for each cluster were uploaded individually, using whole genome (hg19 assembly) as the background. To assign genomic regions to genes, all regions from each cluster were uploaded and genomic regions were associated with genes based on a window of 5.0 kb upstream and 1.0 kb downstream of the gene TSS (Basal +extension setting). The resultant gene-region pairings were downloaded. To determine enriched biological programs for each cluster, only those regions that showed a log2fc > 1 when comparing either in vivo time point to in vitro were used. To assess the biological enrichment for only significant peaks, we performed an ANOVA for each individual peak across the three time points. GREAT was then performed again with the pared down list of peaks with $p < 0.05$.

### Motif analysis
Putative cluster-specific transcription-factor regulators were identified through differential motif-mining analysis using GimmeMotifs maelstrom[53]. Peak lists for each cluster were catted together and reformatted according to the two-column input option. Gimme maelstrom was then run with default parameters. Motifs visualized in the heatmap are those that meet a z-score threshold of 6.

### Statistics
Data in Fig. 5 are displayed as mean ± SD. Significance values are calculated by student's t-test. *** <0.0005, ** <0.005, * <0.05, n.s ≥ 0.05. Confidence intervals and p-values from a two-sided fisher test are

displayed in Supplementary Fig. 2C. *** <0.0005. P-values in figure Supplementary Fig. 2D were generated by a one-sided hypergeometric test.

### Reporting summary
Further information on research design is available in the Nature Portfolio Reporting Summary linked to this article.

## Data availability
Sequencing data generated for this manuscript can be found at the Gene Expression Omnibus (GEO) under the accession number GSE215765. The MNNG-HOS JQ1 RNA-seq data that was previously published is also available on GEO under the accession number GSE74230. Source data are provided with this paper.

## Code availability
All custom code used in this manuscript can be found at the following GitHub repository: https://github.com/wdpontius/temporal_chromatin_accessibility_osteo[54].

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

## Acknowledgements

This research was supported in part by grants from the National Institutes of Health R01CA193677 (P.C.S.), T32GM088088 (W.D.P.), F31CA247266 (W.D.P.), F31CA213965 (I.M.B.), and T32GM007250 (E.S.H., D.H.C). We thank members of the Scacheri lab and Tesar lab past and present for their conceptual and technical feedback. We thank Dr. Stevephen Hung for helpful discussion regarding interpretation and early-stage writing and Dr. Thomas LaFramboise for his insight on statistical analyses. We also thank Drs. Michelle Longworth, Yan Li, Jacob Scott, and Yogen Saunthararajah for their advice regarding this work. Additional support was provided by the CWRU Genomics Core Facility, the CWRU Small Molecule Drug Development Core Facility, the CWRU School of Medicine Department of Genetics and Genome Sciences, and the CWRU School of Medicine Department of Molecular Medicine.

## Author contributions

W.D.P. and P.C.S. conceived of the project and designed the experiments. Z.J.F. designed and prepared the CRISPR screen library. W.D.P. performed the CRISPR screen experiments and analyzed the data. W.D.P. generated and analyzed the ATAC-seq and RNA-seq data. E.S.H. performed and analyzed the scATAC-seq experiments. W.D.P. and I.B. performed and analyzed the ex vivo metastasis experiments. B.E.G. and D.H.C. analyzed and visualized the patient RNA-seq data. C.F.B. performed the ChIP-seq experiments, and K.L. analyzed the data. J.G. and C.D.P. assisted with experimentation. W.D.P. and P.C.S. wrote the manuscript. All authors edited and approved the final manuscript.

## Competing interests

The authors declare no competing interests.
