## [Peer review file · Nature Communications]

REVIEWER COMMENTS

Reviewer #1 (Remarks to the Author):

In this manuscript “Temporal chromatin accessibility changes define transcriptional states essential for osteosarcoma metastasis”, Pontius and colleagues endeavor to understand the mechanisms responsible for metastatic spread to the lung in osteosarcoma. This is an important problem as mechanisms of metastatic spread in general are poorly understood. They describe temporal epigenetic states during metastatic colonization using an intravenous model with GFP-labelled osteosarcoma cell lines. They integrated chromatin accessibility and gene expression data to define master regulators involved in early/late stages of colonization of the lungs, as well as the differences between an in vitro and in vivo models. They performed the assays using MG63.3 cell line generated by in vivo passage, and two HOS derivatives, 143B-HOS (an OS cell line expressing oncogenic K-Ras) and MNNG-HOS transformed with 0.01 mg/mL N-methyl-N0-nitro-N-nitrosoguanidine (MNNG).

The novelty of this work is that they describe the dynamic changes of the epigenetic landscape in the adaptation process of osteosarcoma cells in a distal organ, uncovering a distinct set of transcription factors regulating these temporal states at an early and late stages of lung colonization. Combined use of RNAseq and ATACseq is also a strength as it allows them to uncover the underlying transcriptional networks. They identified 8 ATAC clusters showing the dynamic changes of the chromatin accessibility between the three conditions.

The authors further evaluate transcription factors dependencies using an CRISPR screen in vitro and in vivo, emphasizing the importance of the identification of dependencies in the lung microenvironment. They validated two of the in vivo metastatic-specific dependencies (KLF4 and STAT3) using the PuMA system and uncovered both transcription factors are metastasis dependency genes. Moreover, they identified that the tumors cells in the lung microenvironment show a transcriptional addiction involving the transcription factors previously identified and their expression are decreased after the treatment with JQ1.

The manuscript is overall well-written and the study identifies several novel vulnerabilities of osteosarcoma cells in the lung microenvironment, highlighting the importance of the exploration of early time points in lung metastasis and emphasizing the context-specific importance with the caveat that the study was performed in only two cell lines. This study makes an important contribution to the epigenetic landscape in the temporal states of osteosarcoma cells during metastasis. While the manuscript is suitable for publication in Nature Communications, the limited number of cell lines used potentially diminishes the broad relevance of these findings. An additional limitation is the use of the intravenous metastasis model. To mitigate these issues, it would be useful to perhaps determine whether any of the findings reported here have clinical relevance to osteosarcoma using public databases. Is expression of any of the genes relevant to survival in OS? While it is appreciated that many of these genes may only be upregulated in vivo in early stages of metastasis, it would still be of use to at least test whether any of them could be used as biomarkers to predict metastatic risk.

Major points

1. The authors describe the epigenetic landscape during temporal states of metastasis. They focused their studies primarily on two cell lines to describe the epigenetic landscape. One of these (143B) expresses oncogenic KRAS which is extremely uncommon in OS. The authors should comment in the discussion regarding the limitations of doing these studies in only 2 OS cell lines given the well-known heterogeneity of this disease. More importantly, little effort is made in this manuscript to place these findings in the broader context of OS biology. For example, of the identified TFs, are they generally more highly expressed in RNAseq of metastatic tumors? The authors argue that perhaps these TFs are only upregulated early in the metastatic cascade. Nevertheless, evaluating their expression in public datasets would enrich this manuscript and potentially increase its relevance to our understanding of metastasis in the human disease.

2. The authors claimed differences in gene expression in the three evaluated conditions. However, the differences in the plot shown in Fig. 2, panel C are not appreciated. Why are the differences in the ATAC scores so much more evident than they are for RNAseq? As presented, the idea that gene expression profiles change in the three conditions is not convincing.

3. Most of the atac-seq clusters were also found in the second cell line HOS-143b. The authors should describe what is the degree of overlap between the cell lines per cluster and include it in Suppl Fig. 2.

4. They observed that the correlation between accessible chromatin and expression in clusters 5-8 is weak in Fig 1. "This implies the accessible regions in clusters 5 through 8 may include architectural or negative regulatory elements such as insulators/silencers that dominate the control of associated genes". They also observed in the analysis in Fig 3 that most of the peaks are annotated as distal intergenic in these clusters. The authors should state what they used as the window to associate peaks to genes.

5. Please include a brief explanation how the 78 TF were chosen to be included in the screen. A supplemental figure with the TFs chosen and their expression values in the 3 conditions should be included.

6. A weakness of the using human cell lines in NSG mice is the lack of an immune system. While perhaps doing these experiments using syngeneic cell lines is out of the scope of this manuscript, this issue should be mentioned in the discussion.

Minor points

1. Legend figure 1. They referred in vitro as one metastatic timepoint.

2. The labels in Fig 1D are hard to read and not particularly informative.

3. The legend for C) says "in vivo".

4. Some references are missing.

5. Poor quality of some figures such as Fig. 2, panel D. Also Fig. 3B hard to see as very small and colors very light. Authors should review the quality of all the figures.

6. Suppl Fig 2. Number of peaks per cluster is missing.

7. MNNG-HOS is not mentioned in methods.

Reviewer #2 (Remarks to the Author):

In this study, the authors analyzed the dynamic chromatin accessibility and gene expression landscape during the metastatic colonization of osteosarcoma cells in lung. By generating and analyzing time-course ATAC-seq and RNA-seq experiments, they identified a number of transcription factors that may play important roles in the lung colonization of osteosarcoma. A few transcription factors are further validated using genetic and pharmacological inhibition which show that upstream TFs may represent a class of metastasis-specific dependency genes. Compared to the conventional analyses that compare endpoints of the metastasis process, this study tried to show that the time course analysis of the dynamic process of the colonization in the metastatic cascade may reveal previously undetectable therapeutic opportunities. Overall, the manuscript is clearly written and easy to follow. The findings are useful and worthwhile to be documented in the literature. However, there are several issues that need to be addressed.

Major issues:

1. While ATAC-seq and RNA-seq data from multiple mice were generated, there was no statistical evaluation of the variability of chromatin accessibility and gene expression across different mice, making it difficult to interpret the observed differential chromatin or gene expression between time points. Are the observed differences statistically significant after accounting for the variation across mice or can they simply occur due to mouse-to-mouse variation? Which differential genes or peaks remain statistically significant after accounting for such variation? Is the GREAT analysis (e.g. Fig 2C) based on the statistically significant peaks? Note that many software tools (e.g. MACS) do not directly account for such variation. However, consideration of such cross-individual variation is critical in order to ensure the generalizability of the conclusions.
2. Figure 2C: the temporal expression patterns do not clearly correlate with the temporal chromatin patterns shown in Figure 2B. For example, clusters 2, 3, 4, 5, 6 all show similar expression patterns in Fig 2C but they have substantially different chromatin patterns in Fig 2B. It is unclear whether one can draw any definitive conclusion such as "we would expect the expression of their target genes to mirror the changes in accessibility. Analysis ... revealed [this was indeed the case] for the first 4 clusters". I'm not convinced that "this was indeed the case".

3. Figure 2D: it will be helpful if the authors can comment on why cluster 3 is enriched in regulation of lymphocyte and T cell differentiation.

4. The comparison between different osteosarcoma cell line models is an important analysis to reveal conserved or cell-line specific signatures. However, the current comparison is not fully informative. Based on Supp Fig 2A-B, the authors state that "this indicates that even in tumors with genetically diverse backgrounds, [common] transcriptional programs are required to successfully navigate the different stages of lung metastasis represented in our model". It is unclear whether the observed partial overlap can support this claim, because it is not clear whether the overlap between the two cell line models is statistically significant or can be simply explained by chance. Also, in order to understand the global similarity of the transcriptional programs, it will be much more informative to draw a scatterplot where each data point is a gene ontology term or a geneset or a pathway, and the x- and y-coordinates are the enrichment values of each GO term/geneset/pathway in the two cell line models. Using this scatterplot, one can evaluate how similar or correlated the gene ontology enrichment is between the two cell line models and whether the similarity is statistically significant.

5. Similarly in Supp Fig 2C, a scatterplot comparing motif enrichment between the two cell line models will be useful to fully evaluate the degree of conservation.

6. The computer code for the analyses should be released (e.g. via github, etc.) to ensure reproducibility.

7. The number of time points (3) is still very small to study such a complex dynamic process. It should be discussed as a limitation of the study if the authors cannot analyze more time points.

Reviewer #3 (Remarks to the Author):

In their manuscript „Temporal chromatin accessibility changes define transcriptional states essential for osteosarcoma metastasis“, Pontius et al describe the temporal transcriptional and epigenetic changes during the formation of osteosarcoma lung metastases through the analysis of RNA and ATAC sequencing of cell lines intravenously injected into mice and extracted from their lungs at different time points. This is followed by an elegant in vivo CRISPR drop out screen of transcription factors identified as upstream regulators of the transcriptional changes observed. This identifies the AP1 TFs as well as others as metastasis-specific dependencies, eg. KLF4 and STAT3. The authors nicely validate the metastasis-specific dependency of osteosarcoma cells to KLF4 and STAT3 in elaborate metastasis model systems.

This manuscript is nicely written and addresses an important question in the field. It is technically rigorous, and the authors used state-of-the-art metastasis model systems. The analyses of the RNA seq and ATAC seq seem robust and the CRISPR screen is well designed. The results support the authors main conclusions. I very much enjoyed reading this manuscript and strongly recommend its publication.

Minor comments:

1. How do the ATAC and RNA sequencing profiles compare to those found in primary tumors and or cell lines that are not in the lung? Are any of the programs active during metastasis formation also present in primary tumors, i.e. is part of the metastasis formation process similar to the formation of a tumor in the bone?
2. Would the authors have access to RNA sequencing data from primary tumors vs. metastases? If not from osteosarcoma, maybe from other tumors? I am not aware of such a datasets, but if the authors could find such data and test if some of their findings can also be observed in patients and maybe even in other tumor entities, this could further support their conclusions and extend the relevance to lung metastases formed from tumors other than osteosarcoma.
3. Based on the number of metastases observed it seems that many of the injected cells do not metastasize into the lung. Correct? This is outside the scope of this manuscript, but maybe the authors could mention what happens to most of the cells injected and discuss the role of cells' state before they seed the lung. Are most cells lost during this procedure? Do they die in circulation? Do the mice also form other metastases after tail vein injection? If so, it might be interesting to compare the changes in these metastases and compare them to those observed in the lung. Again, this is outside the scope of this manuscript, but may be of interest to the readers.
4. Based on which criteria did the authors decide to validate KLF4 and STAT3?
5. The authors nicely show that the TF dependencies are distinct in cells that form metastasis and those that grow in vitro. Many stressors during metastasis formation such as hypoxia, 3D growth etc. can be modelled in vitro. Have the authors considered modeling some of the known changes that occur during metastasis formation in vitro to see if these changes are sufficient to render cells dependent on the TFs, eg. KLF4 and/or STAT3? I.e. could these dependencies be merely created through the stress that occurs during metastasis formation but not be required for the actual process of metastasis formation? I would be interested to hear the authors thoughts on this and maybe it would be worth discussing this in the text.

Response to reviewers

We thank the reviewers for their helpful and thoughtful comments. We have addressed these points through new analyses and clarifications of the text. We think these changes have improved our manuscript and strengthened our conclusions. We are excited to hear what the reviewers have to think.

Our study uses a combination of temporal profiling and *in vivo* functional genomics to identify essential epigenomic states that osteosarcoma cells traverse to colonize the lung during metastasis. While most studies focus on comparisons between primary tumors and late stage metastatic lesions, we focus on intermediate time points during a single phase of the metastatic cascade (lung colonization) to highlight the dynamic role of the epigenome during processes that are classically depicted as static. In addition to the importance of these findings for the osteosarcoma field, we believe this work has broad implications for metastasis in other cancers and other secondary organs.

The reviewers' questions were consolidated into three main points. Below we summarize the major new findings. Following this summary, we include a point-by-point response to all reviewer questions.

1. Address the technical concerns of all the reviewers.

- a. The technical concerns of all reviewers have been addressed in a point-by-point manner. The responses are included below.

2. Release the computer code as suggested by Reviewer #2.

- a. All custom code used to perform main analyses have been deposited on github at https://github.com/wdpontius/temporal_chromatin_accessibility_osteo following the link: https://github.com/wdpontius/temporal_chromatin_accessibility_osteo. The manuscript has been revised to include this link, providing access to all readers.

3. Strengthen the clinical relevance of your study.

- b. New data has been included assessing the expression of our TF dependency genes in an independent cohort of osteosarcoma primary tumors and metastases (St. Jude hospital). This analysis demonstrated that 3/8 transcription factor dependencies, including the functionally validated TFs KLF4 and STAT3, show an increase in expression in metastases vs. primaries (Figure 7E). These data demonstrate broader applicability of our specific findings outside of the cell lines originally assessed in our manuscript.

Reviewer #1:

In this manuscript "Temporal chromatin accessibility changes define transcriptional states essential for osteosarcoma metastasis", Pontius and colleagues endeavor to understand the mechanisms responsible for metastatic spread to the lung in osteosarcoma. This is an important problem as mechanisms of metastatic spread in general are poorly understood. They describe

temporal epigenetic states during metastatic colonization using an intravenous model with GFP-labelled osteosarcoma cell lines. They integrated chromatin accessibility and gene expression data to define master regulators involved in early/late stages of colonization of the lungs, as well as the differences between an in vitro and in vivo models. They performed the assays using MG63.3 cell line generated by in vivo passage, and two HOS derivatives, 143B-HOS (an OS cell line expressing oncogenic K-Ras) and MNNG-HOS transformed with 0.01 mg/mL N-methyl-N0-nitro-N-nitrosoguanidine (MNNG).

The novelty of this work is that they describe the dynamic changes of the epigenetic landscape in the adaptation process of osteosarcoma cells in a distal organ, uncovering a distinct set of transcription factors regulating these temporal states at an early and late stages of lung colonization. Combined use of RNAseq and ATACseq is also a strength as it allows them to uncover the underlying transcriptional networks. They identified 8 ATAC clusters showing the dynamic changes of the chromatin accessibility between the three conditions.

The authors further evaluate transcription factors dependencies using an CRISPR screen in vitro and in vivo, emphasizing the importance of the identification of dependencies in the lung microenvironment. They validated two of the in vivo metastatic-specific dependencies (KLF4 and STAT3) using the PuMA system and uncovered both transcription factors are metastasis dependency genes. Moreover, they identified that the tumors cells in the lung microenvironment show a transcriptional addiction involving the transcription factors previously identified and their expression are decreased after the treatment with JQ1.

The manuscript is overall well-written and the study identifies several novel vulnerabilities of osteosarcoma cells in the lung microenvironment, highlighting the importance of the exploration of early time points in lung metastasis and emphasizing the context-specific importance with the caveat that the study was performed in only two cell lines. This study makes an important contribution to the epigenetic landscape in the temporal states of osteosarcoma cells during metastasis. While the manuscript is suitable for publication in Nature Communications, the limited number of cell lines used potentially diminishes the broad relevance of these findings. An additional limitation is the use of the intravenous metastasis model. To mitigate these issues, it would be useful to perhaps determine whether any of the findings reported here have clinical relevance to osteosarcoma using public databases. Is expression of any of the genes relevant to survival in OS? While it is appreciated that many of these genes may only be upregulated in vivo in early stages of metastasis, it would still be of use to at least test whether any of them could be used as biomarkers to predict metastatic risk.

We thank reviewer number 1 for their positive feedback and helpful comments. We have addressed the major points made below.

Major points

1. The authors describe the epigenetic landscape during temporal states of metastasis. They focused their studies primarily on two cell lines to describe the epigenetic landscape. One of these (143B) expresses oncogenic KRAS which is extremely uncommon in OS. The authors should comment in the discussion regarding the limitations of doing these studies in only 2 OS

cell lines given the well-known heterogeneity of this disease. More importantly, little effort is made in this manuscript to place these findings in the broader context of OS biology. For example, of the identified TFs, are they generally more highly expressed in RNAseq of metastatic tumors? The authors argue that perhaps these TFs are only upregulated early in the metastatic cascade. Nevertheless, evaluating their expression in public datasets would enrich this manuscript and potentially increase its relevance to our understanding of metastasis in the human disease.

We have revised the text to more explicitly state the limitations of our work due to our focus on two cell lines. We have also performed a new analysis regarding the expression of key transcription factors identified from our work in osteosarcoma patient primary tumors and metastases (Figure 7E). This analysis showed that out of the 8 *in vivo* dependencies characterized, 3 are overexpressed in metastases, strengthening the application of our findings to osteosarcoma in patients. In addition, a key late-state mediator (SOX9) was overexpressed in these late metastatic lesions. While we would have liked to perform this analysis in a larger cohort of patient samples, the vast majority of data sets available focus on primary tumors, with clinical annotation of whether or not the cancer eventually metastasized. Since the dynamic changes we characterize only emerge in the context of lung colonization, we *do not* expect expression levels of these factors in primary tumors to correlate with eventual metastatic progression.

2. The authors claimed differences in gene expression in the three evaluated conditions. However, the differences in the plot shown in Fig. 2, panel C are not appreciated. Why are the differences in the ATAC scores so much more evident than they are for RNAseq? As presented, the idea that gene expression profiles change in the three conditions is not convincing.

We agree with the reviewer that the differences in RNA expression are much more modest than the ATAC-seq scores. We think this is due to noise associated with looking at all genes that are associated with ATAC-seq peaks, as multiple genes are often associated with a single peak. We reevaluated this analysis focusing on GREAT term-associated genes for each ATAC-seq cluster. In doing this we see that the association between gene expression and chromatin dynamics is mirrored much closer, reinforcing the idea that chromatin changes do positively regulate biology

critical for responding to the metastatic timepoints we assess.

3. Most of the atac-seq clusters were also found in the second cell line HOS-143b. The authors should describe what is the degree of overlap between the cell lines per cluster and include it in Suppl Fig. 2.

A formal statistical analysis evaluating the degree of overlap between cell lines per cluster has been included in Supp Fig 2. In this analysis we used a fisher test to determine if the overlap between equivalent clusters in each cell line was greater than would be expected by chance. To do this, the peak sets included in the contingency table were generated using bedtools intersect. The peak sets that make up the contingency table can be described as follows:

1. MG63.3 specific dynamic peaks for cluster x
2. 143b-specific dynamic peaks for cluster x
3. Overlapping cluster x peaks
4. Other peaks within the universe of 143b and MG63.3 peaks.

The overlaps between equivalent clusters were significantly greater than would be expected by chance, with high odds ratios (>1). This analysis does not change the conclusions made previously. We thank the reviewer for this suggestion as it increases the robustness of our findings.

4. They observed that the correlation between accessible chromatin and expression in clusters 5-8 is weak in Fig 1. “This implies the accessible regions in clusters 5 through 8 may include architectural or negative regulatory elements such as insulators/silencers that dominate the control of associated genes”. They also observed in the analysis in Fig 3 that most of the peaks are annotated as distal intergenic in these clusters. The authors should state what they used as the window to associate peaks to genes.

We have included in the text information about the window used to associate peaks to genes. For reference, the genomic regions were associated with genes based on a window of 5.0 kb upstream and 1.0 kb downstream of the gene TSS.

5. Please include a brief explanation how the 78 TF were chosen to be included in the screen. A supplemental figure with the TFs chosen and their expression values in the 3 conditions should be included.

The library of 78 TFs was curated based on motif enrichment in metastasis-specific enhancers in the MG63.3 cell line. A supplemental figure including their expression across all three conditions has been included (Supp Fig 6).

6. A weakness of the using human cell lines in NSG mice is the lack of an immune system. While perhaps doing these experiments using syngeneic cell lines is out of the scope of this manuscript, this issue should be mentioned in the discussion.

We thank reviewer 1 for acknowledging that experiments in syngeneic cell lines and immune competent mice are outside of the scope of this manuscript. We agree this is a limitation to this current work, and have addressed this in the text.

Minor points

1. Legend figure 1. They referred in vitro as one metastatic timepoint.

The legend was changed to read "*in vitro* and two metastasis timepoints".

2. The labels in Fig 1D are hard to read and not particularly informative.

It is unclear what this refers to as Fig 1 is composed of panels A-C. However, we have taken care to reassess all figures and make sure the labels are legible and informative.

3. The legend for C) says "in vivo".

The legend for Supp Fig 1C was changed to just read "ATAC-seq data sets".

4. Some references are missing.

Missing references were added.

5. Poor quality of some figures such as Fig. 2, panel D. Also Fig. 3B hard to see as very small and colors very light. Authors should review the quality of all the figures.

The poor quality of Fig. 2 is due to compression of the figure to meet the size requirements for initial submission. The quality of the image will be high when the full vector image is submitted for final publication.

For Fig. 3B, the goal of the light color for some genes was to highlight the specific genes that show expression patterns mimicking the accessibility of their respective cluster. We understand that this was unclear and have changed the method of highlighting genes accordingly.

6. Suppl Fig 2. Number of peaks per cluster is missing.

The number of peaks within each cluster was added to Supp Fig 2A.

7. MNNG-HOS is not mentioned in methods.

Since the data from MNNG-HOS was a reanalysis of our previously published dataset, we did not include the cell line in the methods. Clarification of this was added to the methods of this manuscript.

Reviewer #2:

In this study, the authors analyzed the dynamic chromatin accessibility and gene expression landscape during the metastatic colonization of osteosarcoma cells in lung. By generating and analyzing time-course ATAC-seq and RNA-seq experiments, they identified a number of transcription factors that may play important roles in the lung colonization of osteosarcoma. A few transcription factors are further validated using genetic and pharmacological inhibition which show that upstream TFs may represent a class of metastasis-specific dependency genes. Compared to the conventional analyses that compare endpoints of the metastasis process, this study tried to show that the time course analysis of the dynamic process of the colonization in the metastatic cascade may reveal previously undetectable therapeutic opportunities. Overall,

the manuscript is clearly written and easy to follow. The findings are useful and worthwhile to be documented in the literature. However, there are several issues that need to be addressed.

We thank reviewer 2 for their positive comments.

Major issues:

1. While ATAC-seq and RNA-seq data from multiple mice were generated, there was no statistical evaluation of the variability of chromatin accessibility and gene expression across different mice, making it difficult to interpret the observed differential chromatin or gene expression between time points. Are the observed differences statistically significant after accounting for the variation across mice or can they simply occur due to mouse-to-mouse variation? Which differential genes or peaks remain statistically significant after accounting for such variation? Is the GREAT analysis (e.g. Fig 2C) based on the statistically significant peaks? Note that many software tools (e.g. MACS) do not directly account for such variation. However, consideration of such cross-individual variation is critical in order to ensure the generalizability of the conclusions.

While multiple mice were used for each time point to generate the ATAC-seq and RNA-seq data, reviewer 2 is correct in stating no statistical evaluation of variability is made. However, we did perform principal component analysis on omics data collected from all mice and showed the different timepoints cluster separately while different mice at each given timepoint cluster together (Figure 1B). We used this observation as evidence that the dynamic changes we defined between conditions were not due to mouse-to-mouse variability. We have changed the figure legend to more explicitly state that each point on the PCA represents data collected from a different replicate.

For the GREAT analysis, we used peaks where the fold change was greater than 2 when comparing the average ATAC-seq signal at one of the *in vivo* timepoints vs. the *in vitro* timepoint.

2. Figure 2C: the temporal expression patterns do not clearly correlate with the temporal chromatin patterns shown in Figure 2B. For example, clusters 2, 3, 4, 5, 6 all show similar expression patterns in Fig 2C but they have substantially different chromatin patterns in Fig 2B. It is unclear whether one can draw any definitive conclusion such as "we would expect the expression of their target genes to mirror the changes in accessibility. Analysis ... revealed [this was indeed the case] for the first 4 clusters". I'm not convinced that "this was indeed the case".

We agree with the reviewer that the differences in RNA expression are much more modest than the ATAC-seq scores. We think this is due to noise associated with looking at all genes that are associated with ATAC-seq peaks, as multiple genes are often associated with a single peak. We reevaluated this analysis focusing GREAT term-associated genes for each ATAC-seq cluster. In doing this we see that the association between gene expression and chromatin dynamics is

mirrored much closer, reinforcing the idea that chromatin changes do positively regulate biology critical for responding to the metastatic timepoints we assess.

3. Figure 2D: it will be helpful if the authors can comment on why cluster 3 is enriched in regulation of lymphocyte and T cell differentiation.

Although the model we use lacks functional lymphocytes, we think the ability to evade an immune response during lung colonization would be critical to successful progression. In addition to the terms mentioned by the reviewer, the GREAT analysis reveals differentiation of regulatory T cells as an enriched term. Osteosarcoma cells may possess an intrinsic ability to remodel their chromatin to activate Treg-mediated immune-suppressive programs, regardless of whether a functional immune system is present. This idea has been elaborated on in the text. Future studies validating this concept in syngeneic models would be important.

4. The comparison between different osteosarcoma cell line models is an important analysis to reveal conserved or cell-line specific signatures. However, the current comparison is not fully informative. Based on Supp Fig 2A-B, the authors state that "this indicates that even in tumors with genetically diverse backgrounds, [common] transcriptional programs are required to successfully navigate the different stages of lung metastasis represented in our model". It is unclear whether the observed partial overlap can support this claim, because it is not clear whether the overlap between the two cell line models is statistically significant or can be simply explained by chance. Also, in order to understand the global similarity of the transcriptional programs, it will be much more informative to draw a scatterplot where each data point is a gene ontology term or a geneset or a pathway, and the x- and y-coordinates are the enrichment values of each GO term/geneset/pathway in the two cell line models. Using this scatterplot, one can evaluate how similar or correlated the gene ontology enrichment is between the two cell line models and whether the similarity is statistically significant.

We thank the reviewer for this suggestion and have updated the manuscript to include a more detailed comparison of the enriched transcriptional programs found in each cell line (Added to Supp Fig 2).

To answer this question, we assessed whether the number of overlapping significant GO terms for each cluster was greater than expected by chance. We did this by performing a

hypergeometric test with a threshold for significant GO terms of hyper fdr q-value < 0.005. All clusters showed a significant overlap at $p < 0.05$. This reinforces the idea that the dynamic accessible regions characterized in each cell line converge on partially overlapping biology required for the different phases of lung colonization.

5. Similarly in Supp Fig 2C, a scatterplot comparing motif enrichment between the two cell line models will be useful to fully evaluate the degree of conservation.

We thank the reviewer for this suggestion and have updated the manuscript with the requested figure (Supp Fig 3). There is a high degree of overlap between the enriched motifs found in equivalent clusters for each cell line. Although the correlation value is weak for some of the clusters when looking at *all* motifs analyzed, the r value increases greatly when looking at significantly enriched motifs in each cell line. This analysis indicates despite different genomic regions being included in each dynamic cluster, there is convergence on common upstream mediators.

6. The computer code for the analyses should be released (e.g. via github, etc.) to ensure reproducibility.

The reviewer's request for releasing the code used for analyses has been completed and can be found at the following github link: https://github.com/wdpontius/temporal_chromatin_accessibility_osteo

7. The number of time points (3) is still very small to study such a complex dynamic process. It should be discussed as a limitation of the study if the authors cannot analyze more time points.

We agree with the reviewer that the number of time points analyzed is a limitation to the study. We have added a section titled "Limitations of Study" where we discuss this in more depth.

Reviewer #3 (Remarks to the Author):

In their manuscript „Temporal chromatin accessibility changes define transcriptional states essential for osteosarcoma metastasis“, Pontius et al describe the temporal transcriptional and epigenetic changes during the formation of osteosarcoma lung metastases through the analysis of RNA and ATAC sequencing of cell lines intravenously injected into mice and extracted from their lungs at different time points. This is followed by an elegant in vivo CRISPR drop out screen of transcription factors identified as upstream regulators of the transcriptional changes observed. This identifies the AP1 TFs as well as others as metastasis-specific dependencies, eg. KLF4 and STAT3. The authors nicely validate the metastasis-specific dependency of osteosarcoma cells to KLF4 and STAT3 in elaborate metastasis model systems.

This manuscript is nicely written and addresses an important question in the field. It is technically rigorous, and the authors used state-of-the art metastasis model systems. The analyses of the RNA seq and ATAC seq seem robust and the CRISPR screen is well designed. The results support the authors main conclusions. I very much enjoyed reading this manuscript and strongly recommend its publication.

We thank reviewer 3 for their positive comments and are glad they enjoyed reading our work.

Minor comments:

1. How do the ATAC and RNA sequencing profiles compare to those found in primary tumors and or cell lines that are not in the lung? Are any of the programs active during metastasis formation also present in primary tumors, i.e. is part of the metastasis formation process similar to the formation of a tumor in the bone?

Many of the regions we identify as having dynamic chromatin accessibility in the lung are already open to some extent when outside the lung (i.e. *in vitro*). We see these regions increase in accessibility to varying extents at different timepoints during lung colonization. As we do not have a similar comparison for primary tumors, it is hard to say whether these same regions show dynamic accessibility changes at different times during tumor formation/progression. We thank the reviewer for this interesting question and will keep it in mind for future studies.

2. Would the authors have access to RNA sequencing data from primary tumors vs. metastases? If not from osteosarcoma, maybe from other tumors? I am not aware of such a datasets, but if the authors could find such data and test if some of their findings can also be observed in patients and maybe even in other tumor entities, this could further support their conclusions and extend the relevance to lung metastases formed from tumors other than osteosarcoma.

We agree with the reviewer that RNA sequencing from primary tumors and metastases would allow us to further validate the relevance of our findings to the disease in patients. We were lucky enough to gain access to such a data set and to assess the expression of our key TFs in osteosarcoma primary tumors and lung metastases. This analysis revealed that 3/8 *in vivo* dependency genes increase in expression in lung metastases compared to primary tumors (Figure 7E). These factors include our two validated TFs, KLF4 and STAT3, indicating the findings made in our manuscript may translate to at least a subset of human osteosarcomas.

3. Based on the number of metastases observed it seems that many of the injected cells do not metastasize into the lung. Correct? This is outside the scope of this manuscript, but maybe the authors could mention what happens to most of the cells injected and discuss the role of cells' state before they seed the lung. Are most cells lost during this procedure? Do they die in circulation? Do the mice also form other metastases after tail vein injection? If so, it might be interesting to compare the changes in these metastases and compare them to those observed in the lung. Again, this is outside the scope of this manuscript, but may be of interest to the readers.

The recommendation by the reviewer to elaborate on the specific process of metastasis formation in our experimental model is understood and we have updated the manuscript accordingly. While metastasis to other sites after tail vein injection does occur with some osteosarcoma models, it is much rarer than metastasis to the lung (similar to the disease in patients). It would be exciting to determine the differences in chromatin between metastatic sites in future work, and this prospect has been added to the discussion for readers.

4. Based on which criteria did the authors decide to validate KLF4 and STAT3?

KLF4 and STAT3 were chosen for validation due to a variety of reasons. First, KLF4 was chosen initially due to the fact that its motif was enriched at the pan in vivo dynamic regions. This indicated KLF4 was one of the key mediators for dictating the epigenetic state of metastasizing cells. By validating this in vivo-specific essentiality, we showed the functional importance of the dynamic chromatin regions acting downstream of this factor. We then turned to STAT3 as another target that could be validated pharmacologically, as KLF4 (and the other interesting hits from our screen) lacks chemical probes/inhibitors. This experiment served two major purposes: 1) to validate another in vivo dependency, broadening the impact of our work past just KLF4, and 2) to show that our findings had therapeutic implications by identifying specific molecular targets that could be inhibited with small molecules to prevent lung colonization. We thank the reviewer for this comment and have altered the text to make sure these motivations are clear.

5. The authors nicely show that the TF dependencies are distinct in cells that form metastasis and those that grow in vitro. Many stressors during metastasis formation such as hypoxia, 3D growth etc. can be modelled in vitro. Have the authors considered modeling some of the known changes that occur during metastasis formation in vitro to see if these changes are sufficient to render cells dependent on the TFs, eg. KLF4 and/or STAT3? I.e. could these dependencies be merely created through the stress that occurs during metastasis formation but not be required for the actual process of metastasis formation? I would be interested to hear the authors thoughts on this and maybe it would be worth discussing this in the text.

The reviewer makes an excellent point that the microenvironmental pressures faced during metastasis could be isolated and modeled individually *in vitro*. This would be a great way to determine what the major bottlenecks are to lung colonization and allow us to potentially study these individual processes in more detail. While we have not modeled any of these specific stressors *in vitro* to test their relevance to our metastasis findings, we think this would be an interesting future direction. Specifically, we could imagine performing ATAC-seq on cells grown in these different conditions (3D growth, hypoxia, under shear stress, in a transwell with lung fibroblasts). Then, we could see how the open chromatin profiles in these *in vitro* models compared to the metastatic time points we assessed. We anticipate our *in vivo* findings represent a response to a combination of multiple stressors. However, we would expect to see a subset of our dynamic accessible regions appear in some of these proposed *in vitro* conditions. We could then test whether any of our TF dependencies compromise growth in these models. We have updated the manuscript to briefly discuss this exciting direction.

REVIEWER COMMENTS

Reviewer #1 (Remarks to the Author):

The authors have addressed the comments of all three reviewers in a thoughtful and comprehensive manner. I have no additional concerns and I would recommend publication of this manuscript as I believe it is of broad interest to the readers of nature communications and makes an important contribution to the field.

Reviewer #2 (Remarks to the Author):

The authors have addressed some of my questions, but a number of questions still remain. Based on their responses, I recommend the authors consult with a statistician to fully address these concerns in order to ensure the statistical rigor of the study.

1. Regarding my question 1: "Are the observed differences statistically significant after accounting for the variation across mice or can they simply occur due to mouse-to-mouse variation? ... Which differential genes or peaks remain statistically significant after accounting for such variation?...":

The principal component analysis only shows that global chromatin and transcriptional profiles from mice at the same time point are more similar than those from different time points. The PCA does not tell one whether the within-time-point variation is smaller than the between-time-point variation for each individual peak. As a hypothetical example, there may be only 200 true differential peaks and they can provide enough information to separate different time points in the PCA plot, but your analysis may have reported 20000 differential peaks, so most (99%) of the reported peaks are false discoveries and the downstream analyses based on these peaks would be problematic. Therefore, my question 1 was not addressed by the authors.

2. Regarding my question 2: "Figure 2C: the temporal expression patterns do not clearly correlate with the temporal chromatin patterns shown in Figure 2B."

My concern still remains. First, I don't know why the new figure in the response letter is not added to the manuscript. Second, in this new figure, cluster 1-4 showed similar expression patterns, but the chromatin patterns of these four clusters are very different. Third, for the Fig 2B-C in the revised manuscript, the expression patterns for clusters 2-5 all look similar but their chromatin changes are very different. For cluster 3, the expression at the early timepoint is actually lower than the in vitro. For cluster 4, the expression at the early timepoint is slightly higher than in vitro, but the chromatin accessibility at the early timepoint is lower than the in vitro. Based on the data presented in the response letter and in the revised manuscript, I do not believe that the claim in Line 138-142 (i.e. "expression of their target genes to mirror the changes in accessibility" and "this was indeed the case for the first 4 clusters (Fig 2C)") is justified.

3. Line 117: "significant overlap (Supp Fig 1C) - provide the name of statistical test and p-value to justify "significant".

Reviewer #3 (Remarks to the Author):

All my comments were fully addressed by the authors. I would like to congratulate the authors on this manuscript, which I enjoyed reading. Anton Henssen

Response to reviewers

We thank the reviewers for their second round of comments. We think we have addressed the remaining concerns with additional analyses or changes in the text and hope the reviewers agree.

Reviewer #1 (Remarks to the Author):

The authors have addressed the comments of all three reviewers in a thoughtful and comprehensive manner. I have no additional concerns and I would recommend publication of this manuscript as I believe it is of broad interest to the readers of nature communications and makes an important contribution to the field.

We thank reviewer 1 for their kind comment, and for their insight throughout the review process.

Reviewer #2 (Remarks to the Author):

The authors have addressed some of my questions, but a number of questions still remain. Based on their responses, I recommend the authors consult with a statistician to fully address these concerns in order to ensure the statistical rigor of the study.

1. Regarding my question 1: "Are the observed differences statistically significant after accounting for the variation across mice or can they simply occur due to mouse-to-mouse variation? ... Which differential genes or peaks remain statistically significant after accounting for such variation?...":

The principal component analysis only shows that global chromatin and transcriptional profiles from mice at the same time point are more similar than those from different time points. The PCA does not tell one whether the within-time-point variation is smaller than the between-time-point variation for each individual peak. As a hypothetical example, there may be only 200 true differential peaks and they can provide enough information to separate different time points in the PCA plot, but your analysis may have reported 20000 differential peaks, so most (99%) of the reported peaks are false discoveries and the downstream analyses based on these peaks would be problematic. Therefore, my question 1 was not addressed by the authors.

We apologize for misunderstanding the previous point, and agree that the PCA does not indicate whether the variability between timepoints is greater than the variability within timepoints for any given peak. After consulting with Dr. Thomas LaFramboise, a biostatistician in our department, we decided to address this point by performing a one-way ANOVA for every genomic region characterized. This allowed us to compare the variance within timepoints (intra-group variance) to the variance between timepoints (inter-group variance) and determine which genomic regions actually show significantly different means across the three different timepoints. Out of the 43,568 peaks used in our GREAT analysis, 1,494 (3.4%) of the peaks had a p-value greater than or equal to 0.05 (code available on github). Redoing the GREAT analysis with only the 43,568 significant peaks yielded the same results present in our manuscript, indicating the validity of our previous analysis. We thank the reviewer for this suggestion, as it increases our confidence in the results.

2. Regarding my question 2: "Figure 2C: the temporal expression patterns do not clearly correlate with the temporal chromatin patterns shown in Figure 2B."

My concern still remains. First, I don't know why the new figure in the response letter is not added to the manuscript. Second, in this new figure, cluster 1-4 showed similar expression patterns, but the chromatin patterns of these four clusters are very different. Third, for the Fig 2B-C in the revised manuscript, the expression patterns for clusters 2-5 all look similar but their chromatin changes are very different. For cluster 3, the expression at the early timepoint is actually lower than the in vitro. For cluster 4, the expression at the early timepoint is slightly higher than in vitro, but the chromatin accessibility at the early timepoint is lower than the in vitro. Based on the data presented in the response letter and in the revised manuscript, I do not believe that the claim in Line 138-142 (i.e. "expression of their target genes to mirror the changes in accessibility" and "this was indeed the case for the first 4 clusters (Fig 2C)") is justified.

We appreciate the reviewer's comment that the chromatin accessibility and gene expression changes do not match up perfectly, perhaps making the claim in Line 138-142 an overgeneralization. We have changed the text to better reflect our observations. In addition, the figure in our previous response letter has been added as figure 2E.

We do, however, maintain our assertion that there is a good correlation between chromatin accessibility and gene expression across the 8 clusters, with the exception of cluster 1 for the newly added figure. We think it is important to note that although we think these accessible regions are acting as active regulatory elements in many cases, the effect that a particular element might have on its associated gene's expression is not necessarily going to be of the same magnitude as the change in accessibility. In addition, regions of chromatin that are accessible at a given time are not necessarily active but may be poised instead. This means accessible regions still provide information about genes that are functional in a specific biological context (metastasis) despite not regulating that gene's expression at the time that the chromatin was profiled. This is all to say the patterns in gene expression and accessibility are unlikely to reflect each other perfectly, although our analyses show they correlate well. We have added these discussion points to the text.

We hope we have addressed the reviewer's concerns and that they agree the main point of our manuscript – that dynamic changes in chromatin accessibility occur during lung metastasis and are critical for colonization of the metastatic microenvironment – is well supported by our data.

3. Line 117: "significant overlap (Supp Fig 1C) - provide the name of statistical test and p-value to justify "significant".

We have changed the text to say "there was some overlap".

Reviewer #3 (Remarks to the Author):

All my comments were fully addressed by the authors. I would like to congratulate the authors on this manuscript, which I enjoyed reading. Anton Henssen

We are grateful to reviewer 3 (Dr. Henssen) for their generous remarks and valuable advice during the review.

REVIEWERS' COMMENTS

Reviewer #2 (Remarks to the Author):

For authors' response "Out of the 43,568 peaks used in our GREAT analysis, 1,494 (3.4%) of the peaks had a p-value greater than or equal to 0.05 (code available on github). Redoing the GREAT analysis with only the 43,568 significant peaks yielded the same results present in our manuscript, indicating the validity of our previous analysis":

(1) Please clarify whether "redoing the GREAT analysis with only the 43,568 significant peaks" is a typo or not? Should it be $43568 - 1494 = 42074$ significant peaks that are used for GREAT? I'm confused here by the authors' text and want to know whether the number of significant differential peaks is 43568, 42074, or 1494? These numbers do matter and can yield completely different interpretations of the validity of the reported findings. I won't be able to make my publication recommendation without knowing which number is correct.

(2) I did not see this important analysis protocol described in the manuscript. Please add this information to methods or somewhere in the manuscript.

My other questions are addressed satisfactorily.

Reviewer #2 (Remarks to the Author):

For authors' response "Out of the 43,568 peaks used in our GREAT analysis, 1,494 (3.4%) of the peaks had a p-value greater than or equal to 0.05 (code available on github). Redoing the GREAT analysis with only the 43,568 significant peaks yielded the same results present in our manuscript, indicating the validity of our previous analysis":

(1) Please clarify whether "redoing the GREAT analysis with only the 43,568 significant peaks" is a typo or not? Should it be $43568 - 1494 = 42074$ significant peaks that are used for GREAT? I'm confused here by the authors' text and want to know whether the number of significant differential peaks is 43568, 42074, or 1494? These numbers do matter and can yield completely different interpretations of the validity of the reported findings. I won't be able to make my publication recommendation without knowing which number is correct.

We thank reviewer #2 for pointing out this discrepancy. They are correct that this is a typo. The response should read, "redoing the GREAT analysis with only the 42,074 significant peaks", indicating the results from our initial analysis remain the same after filtering for significant peaks.

(2) I did not see this important analysis protocol described in the manuscript. Please add this information to methods or somewhere in the manuscript.

We thank reviewer #2 for their response, as well as their suggestion to perform this analysis. We have added a sentence describing these results to the text, as well as the protocol for performing the analysis to the methods and github page.